# A three-dimensional palaeohydrogeological reconstruction of the groundwater salinity distribution in the Nile Delta Aquifer

Joeri van Engelen[1,2], Jarno Verkaik[1,2], Jude King[1,2], Eman R. Nofal[3], Marc F.P. Bierkens[1,2], Gualbert H.P. Oude Essink[1,2]

[1]Department of Physical Geography, Utrecht University, Utrecht, 3584 CB, The Netherlands
[2]Unit Subsurface and Groundwater systems, Deltares, Utrecht, 3584 BK, The Netherlands
[3]Research Institute for Groundwater, National Water Research Center (El-Kanater El-Khairiya), Egypt

*Correspondence to*: Joeri van Engelen (joeri.vanengelen@deltares.nl)

**Abstract.**

Holocene marine transgressions are often put forward to explain observed groundwater salinities that extend far inland in deltas. This hypothesis was also proposed in the literature to explain the large land-inward extent of saline groundwater in the Nile Delta. The groundwater models previously built for the area used very large dispersivities to reconstruct this saline and brackish groundwater zone. However, this approach cannot explain the observed freshening of this zone. Here, we investigated physical plausibility of the Holocene-transgression hypothesis to explain observed salinities by conducting a palaeohydrogeological reconstruction of groundwater salinity for the last 32 ka with a complex 3D variable-density groundwater flow model, using a state-of-the-art version of the computer code SEAWAT that allows for parallel computation. Several scenarios with different lithologies and hypersaline groundwater provenances were simulated, of which five were selected that showed the best match against observations. Amongst these selections, total freshwater volumes varied strongly, ranging from 1526 to 2659 $km^3$, mainly due to uncertainties in the lithology offshore and at larger depths. This range is smaller (1511-1989 $km^3$) when we only consider the volumes of onshore fresh groundwater within 300 m depth. Regardless of this variation, in all cases the total volume of hypersaline groundwater exceeded that of sea water. We also show that during the last 32 ka, the total freshwater volumes significantly declined, with a factor ranging from 1.9 to 5.4, due to the rising sea-level. Furthermore, the time period required to reach a steady state under current boundary conditions exceeded 5.5 ka for all scenarios. Finally, under highly permeable conditions the marine transgression simulated with the palaeohydrogeological reconstruction led to a steeper fresh-salt interface compared to its steady-state equivalent, while low permeable clay layers allowed for the preservation of volumes of fresh groundwater. This shows that long-term transient simulations are needed when estimating present-day fresh-salt groundwater distributions in large deltas. The insights of this study are also applicable to other major deltaic areas, since many also experienced a Holocene marine transgression.

## 1 Introduction

Palaeohydrogeological conditions have influenced groundwater quality in the majority of large-scale groundwater systems (Edmunds, 2001; Jasechko et al., 2017). These conditions can especially be found in deltaic areas, where the effects of marine transgressions are often still observed in groundwater salinities (Larsen et al., 2017). More specifically, their low elevation allowed for far reaching marine transgressions, leading to a large vertical influx of sea water, and hampered subsequent flushing with fresh water after the marine regression. This hypothesis is supported by hydrogeochemical research in several deltas (e.g. Colombani et al., 2017; Fass et al., 2007; Faye et al., 2005; Manzano et al., 2001; Wang and Jiao, 2012). The physical justification for this hypothesis, however, often still has to be tested, with a few notable exceptions (Delsman et al., 2014; Larsen et al., 2017; Van Pham et al., 2019), and can be provided by palaeohydrogeologic modelling, which has recently provided important insights for several cases. Gossel et al. (2010) created a large-scale 3D variable-density groundwater model of the Nubian Aquifer System and showed that seawater intrusion has occurred since the Pleistocene Lowstand towards the Qattara Depression (North-West Egypt). Later, Voss and Soliman (2013) showed with a parsimonious 3D model of the same groundwater system that water tables are naturally declining during the Holocene, since they receive limited recharge and are drained into oases or sabkhas. Moreover, the authors used an inventive validation method by comparing the position of discharge areas in the model with a dataset of oases or sabkha locations. Delsman et al. (2014) conducted a detailed palaeohydrogeological reconstruction of the last 8.5 ka over a cross-section in the Netherlands to show that the system has never reached a steady state. They showed that the Holocene transgression caused substantial seawater intrusion, from which the system is still recovering. Using a combination of geophysical data and 2D numerical models, Larsen et al. (2017) showed that during the Holocene transgression sea water preferentially intruded in former river branches in the Red River Delta, Vietnam. Van Pham et al. (2019) showed that most of the fresh groundwater in the Mekong Delta (Vietnam) was likely recharged during the Pleistocene and preserved by the Holocene clay cap. Despite being in a humid climate, recharge to the deeper Mekong Delta groundwater system is very limited and freshwater volumes are still declining naturally.

The Nile Delta is one of the deltas where the marine transgression hypothesis has not been tested by physical analysis yet, despite severe problems with groundwater salinity (section 2.1). This saline groundwater issue has led to a continuous line of research into fresh and saline groundwater occurrences in the Nile Delta Aquifer (NDA). These studies can be divided into hydrogeochemical and groundwater-mechanical studies. The latter subdivision covers all studies focused on groundwater flow and statics, following Strack (1989). Starting with the hydrogeochemical studies, Geirnaert and Laeven (1992) provided the first conceptual palaeohydrological model (dating back to 20 ka). They used groundwater dating to show that shallow groundwater was likely recharged as fresh river water around 3.5 ka. Barrocu and Dahab (2010) extended this conceptual model to up to 180 ka and concluded that the saline groundwater in the north is old connate groundwater trapped in the Early Holocene, because it is freshening. Geriesh et al. (2015) further supported this conceptual model with additional measurements.

As an example of a study based on groundwater mechanics, Kashef (1983) used the Ghyben-Herzberg relationship to show that the seawater wedge reached far inland, thereby showing that the NDA should be carefully managed despite that the aquifer was gaining fresh water at that time. This was further confirmed by the 2D numerical model constructed by Sherif et al. (1988), who concluded that the width of the dispersion zone may be considerable. Sefelnasr and Sherif (2014) showed the

sensitivity of the area to sea-level rise with a 3D numerical model, as the low topography of the Nile Delta allowed for a far reaching land surface inundation that can be detrimental to freshwater volumes (Ketabchi et al., 2016; Kooi et al., 2000). Mabrouk et al. (2018) showed that groundwater extraction is a larger threat to freshwater volumes than sea-level rise, under the assumption that land surface inundation with sea water is prevented. Van Engelen et al. (2018) investigated the origins of hypersaline groundwater that is found starting from 400 m depth, which was overlooked in previous studies. They tested four

hypotheses of which two remained valid: 1) free convection of hypersaline groundwater formed in the Late-Pleistocene coastal sabkha deposits and 2) upward compaction-induced flow of hypersaline groundwater, formed during the Messinian Salinity Crisis. Moreover, they showed the influence of uncertain lithology on the groundwater salinity distribution.

When comparing the results of the hydrogeochemical studies with those of the groundwater-mechanical studies, a discrepancy can be observed that is difficult to reconcile. All the previously discussed groundwater-mechanical studies

except Van Engelen et al. (2018) neglected the influences of palaeohydrogeological conditions that were inferred by hydrogeochemists. Instead, they used longitudinal dispersivities exceeding 70 m (Sefelnasr and Sherif, 2014; Sherif et al., 1988) to simulate the large brackish zone in the groundwater system (Fig. 1) (Sefelnasr and Sherif, 2014; Sherif et al., 1988). The hydrogeochemical studies, however, attribute this zone to the Holocene transgression, since a large fraction of the brackish zone is presumably freshening (Geriesh et al., 2015; Laeven, 1991) and the extent of this brackish zone coincides

with palaeo-shorelines (Kooi and Groen, 2003; Stanley and Warne, 1993) (Fig. 1). A variable-density groundwater flow model with present-day boundary conditions and a large amount of hydrodynamic dispersion cannot support the freshening of the brackish zone, since hydrodynamic dispersion would only lead to progressive salinization. In addition, currently no reliable field data exists that supports longitudinal dispersivities beyond 10 m (Zech et al., 2015). This discrepancy in conceptualization still exists, despite the large influence of palaeohydrogeologcial conditions on salinities previously shown

by applying variable-density groundwater flow models to other cases (Kooi et al., 2000; Meisler et al., 1984).

The lack of including palaeohydrogeological development in variable-density groundwater flow and coupled salt transport models is understandable as it requires vast amounts of computational resources, especially when 3D reconstructions are considered. The availability of a newly developed model code that allows for high-performance computing (Verkaik et al., 2017), has now made it possible to create the first numerical palaeohydrogeological reconstruction of the Nile Delta Aquifer

in 3D. Therefore, in this study we have constructed a variable-density groundwater flow and coupled salt transport model of the Nile Delta Aquifer and use it for a palaeohydrogeological reconstruction of salinity in the NDA over the last 32 ka. Specifically, we use the model to 1) investigate the physical plausibility of the Holocene transgression hypothesis for the Nile Delta; 2) investigate the influence of the uncertain geology (Enemark et al., 2019); 3) provide volume estimates of

present-day fresh groundwater [FGW]; 4) assess the relative importance of using palaeohydrogeological reconstructions against less expensive steady-state modelling.

## 2 Area description

### 2.1 Area relevance and vulnerability

The Nile Delta is vital for Egypt. It is an important bread basket, as it has been through history (Dermody et al., 2014), since it is the main area suitable for agriculture in an otherwise water-scarce region (WRI, 2008). Furthermore, the area is densely populated (Higgins, 2016), and the population is only expected to increase over the coming decades (World Bank, 2018). Although the area is traditionally irrigated with surface water, agriculture increasingly relies on groundwater (Barrocu and Dahab, 2010). For instance, a majority of interviewed farmers in the central delta indicated that they are pumping

groundwater almost continuously during the summer (El-Agha et al., 2017). This groundwater pumping will increase in the future, as large dams are planned or built upstream, e.g. the Grand Ethiopian Renaissance Dam (GERD), which will reduce the discharge of the Nile and will consequently hamper flushing of the irrigation system. This is expected to further deteriorate the quality of surface water in the irrigation system, which currently already is in a critical state; the water that currently reaches the sea is mostly saline and highly polluted (Stanley and Clemente, 2017). For example, the nitrate

concentrations of the water inflow into the Manzala lagoon have increased by a factor 4.5 in the past 20-25 years, as an effect of the intensified agriculture and the reduced inflow of river water (Rasmussen et al., 2009). Given the increasing stress on the groundwater system, it is important to assess the status of the current and future freshwater volumes.

Despite the large groundwater volume of the Nile Delta Aquifer, the amount of groundwater suitable for domestic, agricultural or industrial water supply is limited. A considerable fraction of the groundwater volume is saline (Kashef, 1983;

Sefelnasr and Sherif, 2014) or even hypersaline (van Engelen et al., 2018), and unsuitable for most uses. Saline groundwater can be detrimental to Egypt's already critical food supply. For example, when increased irrigation caused the level of the brackish groundwater to rise in Tahrir, Cairo, wheat yields reduced by 41% within a four year period as soil salinity increased (Biswas, 1993).

### 2.2 Hydrogeology

The Nile Delta Aquifer (NDA) is very thick, reaching up to 1 km depth at the coast (Sestini, 1989). For reference, only 0.3% of the coastal aquifers in the world are estimated to exceed this thickness (Zamrsky et al., 2018). The NDA consists of the Late-Pliocene El Wastani formation and the Pleistocene Mit Ghamr formation. These both contain mainly coarse sands with a few clay intercalations (Sestini, 1989). It is capped by the Holocene Bilqas formation, which has recently been mapped extensively (Pennington et al., 2017). The Pliocene Kafr El Sheikh formation is generally assumed to be the hydrogeological

base as it is 1.5 km thick and consists mainly of marine clays, though it is possible that there is compaction-driven salt transport through this layer (van Engelen et al., 2018). Hydrogeological research dealing with this area generally assumed

that the connection between aquifer and sea is completely open (e.g. Kashef, 1983; Mabrouk et al., 2018; Sefelnasr and Sherif, 2014; Sherif et al., 1988), but there exists ample evidence against this assumption from both seismics (Abdel-Fattah, 2014; Abdel Aal et al., 2000; Samuel et al., 2003) as well as from the fact that offshore borelogs contain more clay than sands (Salem et al., 2016). These observations can be explained by the fact that the fraction of marine clays generally

increases seawards (Nichols, 2009). These marine clays can even form large vertical clay barriers when deposited during aggradational or retrogradational phases (Nichols, 2009) with a major effect on the groundwater system (van Engelen et al., 2018).

## 2.3 Groundwater salinity

Figure 1 shows all groundwater Total Dissolved Solids (TDS) measurements above 250 m depth that we could find in

literature (Geriesh et al., 2015; Laeven, 1991; Salem and El-Bayumy, 2016), combined with data of the Research Institute for Groundwater (RIGW) (Nofal et al., 2015). Other publications are not included as 1) these did not report measurement depths, 2) did not report measurement locations, or 3) only showed isohalines, from which the actual measurement locations are impossible to discern. The few available salinity measurements deeper than 250 m are not shown here as these are likely not influenced by the Holocene transgression (van Engelen et al., 2018). The depicted measurements highlight considerable

spatial variation, especially in the brackish zone. This variability in measured salinity can be explained with 1) the different measurement depths, 2) different data sources, with different data quality and dates of their measurement campaigns, 3) heterogeneity in the hydraulic conductivity of the subsoil resulting in heterogeneous salt transport, and 4) heterogeneous evapoconcentration. The latter is inferred from the observed salinities that exceed the salinity of sea water (35 g TDS $L^{-1}$), which presumably are pockets of evapoconcentrated Holocene groundwater (Diab et al., 1997). Despite the spatial variability

in salinity, we observe a trend: the extent of the brackish zone seems to conform in the east to the coastline during the maximum transgression at 8 ka. Westwards, the extent of this zone can be explained with the location of the former Maryut lagoon, which had several periods where its salinities approached that of sea water (Flaux et al., 2013).

It is important to note that our dataset is biased towards areas with low soil salinities, as measurements have been preferentially taken in the most productive agricultural areas and thus of higher economic interest to monitor. A comparison

of our dataset with soil salinity maps (Kubota et al., 2017) shows that in the coastal areas soil salinities are high, which are the areas where we have the least measurements. This bias is particularly evident from the data gathered by Salem et al. (2016b). These authors conducted research in the only coastal area with low soil salinities, which is an area that has predominantly consisted of coastal dunes from 7.5 ka until present (Sestini, 1989; Stanley and Warne, 1993), thus providing the hydrogeological circumstances for the development of a freshwater lens. The other, former dune areas (Fig. 3) were

either eroded naturally or removed by humans (El Banna, 2004; Malm and Esmailian, 2013; Stanley and Clemente, 2014) and have high soil salinities, presumably causing former freshwater lenses to salinize. Given this bias in our dataset towards fresh groundwater, the extent of the saline groundwater problem is likely underexpressed in Fig. 1.

# 3 Methods

## 3.1 Lithological model

A dataset of 159 borelogs was compiled using georeferenced data from different literature sources (Coleman et al., 1981; Nofal et al., 2016; Salem et al., 2016; Summerhayes et al., 1978). These borelogs were used to constrain a 3D lithological
5  model using SKUA-GOCAD's implicit modelling engine (Paradigm, 2017). There are two main uncertainties in the lithological model thus constructed. The first pertains to the question to what extent the continental slope is covered with low-permeable clayey sediments (section 2.2) and second whether the clay layers observed in the NDA are continuous, forming low permeable structures, or are disconnected with only a limited effect on regional groundwater flow. To account for these uncertainties, nine different lithological model scenarios were constructed (Fig. 2), where we varied the height of
10  the clayey sediments on the continental slope and the hydraulic conductivity of the onshore-reaching clay layers. The height of the clayey sediments determines how disconnected the deeper groundwater system is from the sea, and thus the ability of the system to preserve denser hypersaline groundwater in its aquifers (van Engelen et al., 2018). The hydraulic conductivity of the onshore-reaching clay layers is varied to get a first-order approximation of the effect of clay layers on regional groundwater flow. We assigned a continuous hydraulic conductivity to these clay layers, based on three different lithologies
15  (in order of decreasing hydraulic conductivity): sand, fluvial clay and marine clay (Table 3). The rationale behind this is that small clay lenses have negligible effect on regional groundwater flow, thus are assigned a hydraulic conductivity of sand. Fluvial clay layers are assigned a hydraulic conductivity of the current confining Holocene clay layer, as this was deposited under fluvial conditions (Pennington et al., 2017). Marine clay layers present continuous layers of low conductivity with a big influence on the regional groundwater flow.

## 3.2 Model code and computational resources

We applied the newly developed iMOD-SEAWAT code (Verkaik et al., 2017) which is based on SEAWAT (Langevin et al., 2008), but supports distributed memory parallelization that allows for a significant reduction in computation times. The SEAWAT code is the industry standard for solving variable-density groundwater and coupled solute transport problems, therefore the reader is referred to its manuals for an extensive explanation (Guo and Langevin, 2002; Langevin et al., 2003, 25  2008). The main improvement of the iMOD-SEAWAT code is that it replaces the original solver packages for variable-density groundwater flow and solute transport (respectively PCG and GCG) with the Parallel Krylov Solver package (PKS). The PKS linear solver is largely based on the unstructured PCGU-solver for MODFLOW-USG (Panday et al., 2013) and solves the global linear system of equations with the additive Schwarz parallel preconditioner (Dolean et al., 2015; Smith et al., 1996) using the Message Passing Interface (The MPI Forum, 1993) to exchange data between subdomains, where each 30  subdomain is assigned its own private memory on a computational node. The variable-density flow problem and the solute transport problem are solved in parallel using respectively the additive Schwarz preconditioned Conjugate Gradient and BiConjugate Gradient Stabilize linear accelerators (Barrett et al., 2006; Golub and Van Loan, 1996). Simulations were

conducted on the Dutch national computational cluster Cartesius (Surfsara, 2014), using Intel Xeon E5-2690 v3 processors. With this new code the model scenarios had a wall clock time ranging from 44 hours to 108 hours on 48 cores, depending on model complexity.

### 3.3 Model domain and numerical discretization

The study area encompasses the complete deltaic plain, the coastal shelf, coastal slope, and the Eastern desert and Western desert fringes (near the Suez Canal and Wadi El Natrun area). Its Western boundary is close to Alexandria, its Eastern the Suez Canal, its Southern near Cairo, and the Northern boundary laid 70 km offshore, resulting in a rectangle of 240 km meridional (W-E) distance and of nearly 260 km zonal (S-N) distance (see also Fig. A1). This area was discretized into 1 by 1 km cells. To determine the vertical dimension, borelogs and bathymetrical/topographical data were used to respectively

determine the bottom and top of the NDA. For the topography of our model, we used a combination of the Shuttle Radar Topography Mission (NASA, 2014) for the onshore part and the General Bathymetric Chart of the Oceans (GEBCO, 2014) for the offshore part. The top of the NDA was clipped off above 20 m AMSL, as the hills above this height have no important contribution to the groundwater flow (Geirnaert and Laeven, 1992), because there is very limited rainfall in the south and no surface water there. The model domain was discretized into 35 model layers, from top to bottom: 21 model

layers of 20 m thickness, 10 model layers of 40 m thickness, and 4 model layers of 50 m thickness. In total, the model has more than 2 million cells. We simulated a time period of 32 ka, covering the Late-Pleistocene and Holocene. This period was chosen as 1) no palaeogeographic maps were available of the preceding period, and 2) this research only focused on the potential effects of the Holocene transgression, not previous Pleistocene transgressions.

### 3.4 Boundary conditions

#### 3.4.1 Stress periods

The palaeohydrogeological reconstruction consisted of several consecutive stress periods (Fig. 3), following Delsman et al. (2014), in which boundary conditions were kept constant. Each stress period was assigned a palaeogeographic map (Pennington et al., 2017; Stanley and Warne, 1993) that defines the location of five geographical classes, namely sea, lagoon,

sabkha, river, and dune/beach. Each was associated with boundary conditions that are explained more extensively in sections 3.4.2–3.4.5; a summary of their data sources can be found in table 2. The lithological classes "clay" and "sand" indicate where the Holocene confining clay layer was respectively located and absent in each stress period. Firstly, stress periods 1 and 2 covered the Late-Pleistocene, where sea-levels were low and therefore the coastline was located ~70 km further to the North and the hydraulic gradient was high. The area consisted of an alluvial plain with braided rivers. In between these

rivers, sabkha (salt flat) deposits were found in local depressions, which stayed fixed in location (Stanley and Warne, 1993). Next, stress period 3 covered the marine transgression, which was a period of rapid sea-level rise at the start of the Holocene. In stress period 4 the delta started prograding in the west and centre, and lagoons were formed. Of particular interest is the

large extent of the Maryut lagoon in the west, filling up what is now known as the Maryut depression (Warne and Stanley, 1993). In stress period5, progradation started in the east, leading to the symmetrical shape in stress period6. The eastwards progradation was caused by the decreased hydraulic gradient that led to finer sediments being deposited, which were then transported eastwards with the dominantly eastwards flowing sea currents. During stress period 7, humans converted most river branches to a system of irrigation channels, leaving only the Rosetta and Damietta river branches to persist.

### 3.4.2 Surface waters

The surface water systems (the sea, rivers, and lagoons), were incorporated as a Robin boundary condition (Jazayeri and Werner, 2019), requiring a specified head, salinity, and bed resistance, the latter being the resistance exerted by the bed sediments on the surface-groundwater interaction. Of these three inputs, the least is known about the bed resistance. A 100 day resistance was used for all surface water systems (Table 3), which is a common value for models of the Rhine-Meuse Delta (De Lange et al., 2014; Timmerman and Hemker, 1993). In Appendix B we discuss the effects of this assumption in more detail.

Sea boundary cells were placed on the edge of the coastal shelf and slope, following the bathymetry (GEBCO, 2014) and, where possible, palaeogeographic maps that specified the coastline (Fig. 3). This meant that for the Late-Pleistocene (stress periods 1 and 2) we had to resort to the present-day bathymetry and the contemporary sea-level to approximate the coastline. We specified the head using palaeo sea-level curves. No specific sea-level curves are available for the Nile Delta (Pennington et al., 2017). Hence we resorted to eustatic sea-levels for the Pleistocene (Spratt and Lisiecki, 2016) and a more local curve for the Holocene (Sivan et al., 2001), as this was the best information available. The minimum sea-level was fixed at -100 m, as below that achieving numerical convergence was cumbersome. This only slightly affected results as land surface inundation at these sea-levels was nearly equal due to the steep coastal slope. The salinity of the sea was set at a constant 35 g TDS L$^{-1}$.

Of the total of seven stress periods, the last four have lagoons, since lagoons started to form from 7.5 ka. The lagoon stage was set such that it was in hydrostatic equilibrium with the sea, so that its pressure is corrected for salinity (Post et al., 2007). Lagoonal palaeo-salinities were estimated from the published strontium isotope ratios from the Maryut lagoon (Flaux et al., 2013) for each stress period. To be specific, the salinities assigned to stress periods 3 to 7 are respectively: 18, 9, 4.5, 2.5 g TDS L$^{-1}$. The decreasing trend in salinities is partly the result of a progressively humid climate and increased human influence (Flaux et al., 2013) and partly the result of averaging over our stress periods. From 4 ka to 3 ka there was an arid period with high salinities (~18 g TDS L$^{-1}$), but since we based our stress periods on the available palaeogeographic maps, this spike is dampened by the preceding 2 ka period of brackish conditions (~5 g TDS L$^{-1}$).

River stages were specified by creating linear profiles from the apex to the coastline. The location of the delta apex and its river stage were fixed through time at the present-day conditions. This is a simplification, as in reality the apex' location varied through time. For instance, the delta apex was located up to 65 km south of its current location somewhere in the last

6 ka (Bunbury, 2013), but due to the lack of data on apex migration for the rest of the modelled time domain we chose this simplification. Since the coastline is irregular in shape and the location of rivers was not fixed through time, river stages were determined in the following manner: Firstly, lines were drawn from the apex to points at the coastline for each raster cell at the coastline. Along these lines, river stages declined linearly to the contemporary sea-level at the coastline. Secondly,

these profiles were subsequently interpolated to a surface using Inverse Distance Weighting. Finally, the contemporary river branches were clipped out of this surface. Rivers were assigned a salinity of 0 g TDS L$^{-1}$, since the Nile Delta is not tide dominated (Galloway, 1975), meaning that saline water intrusion in river branches is limited.

### 3.4.3 Dunes and beaches

Locations where dunes or beaches occurred were assigned a fixed recharge of 200 mm a$^{-1}$, equal to the present-day average

precipitation near Alexandria (WMO, 2006) and also the current recharge along the Levant coast (Yechieli et al., 2010). This is reasonable, as the climate was mainly wetter throughout the Holocene than it currently is (Geirnaert and Laeven, 1992). This recharge allowed for the formation of freshwater lenses underneath dunes and beaches, which were observed in the area during ancient times (Post et al., 2018).

### 3.4.4 Extractions

Groundwater extractions mainly occur in the South-West. This area, near Wadi El Natrun, was appointed a reclamation area by the Egyptian Government in 1990 and since then has seen a rapid increase in extraction rates (King and Salem, 2012; Switzman et al., 2015). This was implemented in the numerical model by including extraction wells (locations and rates) that were in the RIGW database (Nofal et al., 2018) for the last 30 years of the simulation (Fig. A2).

### 3.4.5 Hypersaline groundwater provenances

Below 400 m depth, hypersaline groundwater (HGw) has been observed in the Nile Delta Aquifer. Van Engelen et al. (2018) investigated potential origins of this hypersaline groundwater and identified two potential sources that are included in this model. Firstly, the Late-Pleistocene sabkhas could have been sources of the hypersaline groundwater. Therefore, we set the concentrations of these areas at 120 g TDS L$^{-1}$ for certain scenarios that are specified further in section 3.4.6, so that hypersaline groundwater could be formed in these areas. We refer to these as scenarios where HGw originates from the

"top".

Secondly, another potential source of hypersaline groundwater could be seepage of hypersaline groundwater expulsed from the low-permeable Kafr El Sheikh formation due its compaction. This seepage flux was included in the model as fixed fluxes at the bottom. The magnitude and salt concentration of these fluxes were set at 3E-06 m d$^{-1}$ and 120 g TDS L$^{-1}$ (van Engelen et al., 2018). These scenarios are referred to as those where HGw originates from the "bottom".

## 3.5 Initial salinity distributions

Initial 3D salinity distributions were set with a dynamic spin-up time for 160 ka using Late-Pleistocene boundary conditions. Although the 160 ka period exceeds a glaciation cycle, we observed that no significant changes in salinity concentrations occurred after 4 ka for the homogeneous and 80 ka for the heterogeneous cases, which is well within the range of a glaciation cycle. For the majority of the model scenarios we started the spin-up with a completely fresh Nile Delta Aquifer, except for the model scenarios receiving HGw from the bottom, where the output of a simulation from previous research (model "NDA-c" in van Engelen et al., 2018) was used to set the initial conditions. This was a simulation of the effect of 2.5 Ma of compaction-induced upward flow solute transport through the Kafr El Sheikh formation into the aquifer under interglacial conditions (low hydraulic gradient).

## 3.6 Model scenarios

Van Engelen et al. (2018) identified two sensitive inputs for hydrogeological models in predicting the distribution of hypersaline groundwater, which are the geological model and the source of hypersaline groundwater. Even though there are 18 possible combinations between the nine lithological model scenarios and the two HGw provenances (top and bot), only 13 were calculated. Based on previous findings in Van Engelen (2018), three combinations could be excluded, as these would not produce the observed salinity distributions. These were the combinations with an upward HGw flux originating from the bottom and with a completely open sea boundary, which lead to all HGw to be immediately drained into the sea resulting in only a very small volume of HGw in the NDA. The other two excluded scenarios are those with a closed sea boundary, horizontal clay layers (fluvial or marine), and HGw coming from the top. The free-convective plumes in these scenarios created entrapped volumes of fresh groundwater in between clay layers. Pressures in these zones increased fast during sea-level rise, which, in combination with the heterogeneity, made it impossible to maintain numerical convergence for these two model scenarios, despite our best efforts.

As an addition to the palaeohydrogeological reconstruction, we formulated for each geology-HGw provenance combination, an equivalent steady-state model which used only the present day hydrological forcings until its salinity distribution reached a steady state, comparable to previous 3D models made for the area (Mabrouk et al., 2018; Sefelnasr and Sherif, 2014). This resulted in 26 scenarios in total. To distinguish different model scenarios, we introduced a coding system. For each feature, the corresponding letters in table 1 are converted to a code as follows: *{sea}-{clayer}-{prov}-{temp}*. For example, the palaeohydrogeological reconstruction (*temp*: P) of an aquifer with a half-open sea connection (*sea*: H), fluvial horizontal clay layers (*clayer*: F), and HGw seeping in from the bottom (*prov*: B) gets the following code: H-F-B-P. Its equivalent steady-state model is attributed the code H-F-B-S. Furthermore, all model scenarios that share the same feature are noted as "{feature}-model scenarios", e.g. "T-model scenarios" means all model scenarios where HGw originates from the top, i.e. the Pleistocene sabkhas. The scenarios with a "Homogeneous" lithological model (Fig. 2) get the scenario codes "O-N".

### 3.7 Model evaluation

We compared our modelled salinity distributions to available TDS measurements (n=293), of which the majority is shown in Fig. 1. Comparing measurements with model salinities at the point location can be deceiving as sharp transition zones often occur in salinity distributions (Sanford and Pope, 2010). Therefore, we assessed our model's ability to reproduce measured salinity patterns as follows. First, TDS measurements were binned into classes as shown in Fig. 1, similar to what is common in the validation of hydrogeophysical products (e.g. Delsman et al., 2018). Second, isosurfaces were determined in our model output for all bin edges with the Marching Cubes algorithm (van der Walt et al., 2014). If these classes did not correspond at the measurement location, the minimum displacement ($\Lambda$) to the observed class in the model output was determined by calculating the minimum Euclidean distance in 3D:

$$\Lambda = \min \left| \sqrt{\sum_{i=1}^{3}(o_i - l_i)^2} \ , \ \sqrt{\sum_{i=1}^{3}(o_i - u_i)^2} \right| \tag{1}$$

where $o_i$ is the location of the observation in dimension i and $l_i$ and $u_i$ the locations of the isosurface vertices in dimension i for respectively the lower and upper bin edge (Fig. A3). All locations were normalized to a range of [0,1] since the model domain was a rectangular cuboid, in other words not a perfect cube, and thus unequal in size across each dimension.

To assess the validity of the steady-state assumption, we checked for all equivalent steady-state models the time they reached a steady state. This time was determined by calculating the derivative of the freshwater volume over time. If this did not change more than 1E-04% of the total volume, we considered the model to have reached a steady state.

### 3.8 Comparison palaeohydrogeological reconstruction with its equivalent steady-state model

The palaeohydrogeological reconstruction was compared with its equivalent steady-state model in two manners. Firstly, we calculated:

$$\Delta C = C_\text{p} - C_\text{s} \tag{3}$$

where $C$ is the 3D salinity matrix at the last timestep, and p and s respectively are the palaeohydrogeological and steady-state model scenarios. For locations where $\Delta C > 0$, the palaeohydrogeologocial reconstruction is saltier than its steady-state equivalent and vice versa. Secondly, we calculated isohalines for a set of TDS values (3, 10, 20, 30 g L$^{-1}$) for both model scenarios and assessed at the distance between these isohalines in the *y*-dimension (North-South). As our coastline was irregular, these distances varied in the *x*-dimension (West-East), therefore the median (Md) was calculated across this dimension as a conservative measure of isosurface separation ($\omega$), which for a given depth and concentration is defined as:

$$\omega = \mathrm{Md} \left| \boldsymbol{y}_{\mathrm{p},x} - \boldsymbol{y}_{\mathrm{s},x} \right|_x \tag{4}$$

where $\boldsymbol{y}$ are the isohaline locations in the $y$-dimension, and p and s respectively are the palaeohydrogeological and steady-state model scenarios.

## 4 Results

Five scenarios were selected that showed the best match with the observations. These are called the "acceptable" scenarios. To keep the results comprehensible, we start with discussing the results of these five selected scenarios through space (section 4.1) and time (section 4.2), before discussing this actual selection procedure (section 4.3).

### 4.1 Current spatial TDS distribution of acceptable model scenarios

The acceptable model scenarios show different salinity distributions (Fig. 4), despite similar model performance (Fig. 6). The model scenarios H-F-T-P, H-N-T-P, and C-N-T-P, which have no or fluvial clay layers, have freshwater distributions very similar to the O-N-T-P model scenario. In the model scenarios with marine clay layers (M), however, fresh groundwater is preserved in between low-permeable clay layers, especially in the centre. Regardless of the differences between scenarios, in all realizations the fresh-salt interface roughly follows the coastline, except in the west where there is far extending seawater intrusion visible towards Wadi El Natrun (Fig. 1). Next to this depression, (former) lagoons are visible as shallow brackish zones and (former) dune areas are visible as freshwater lenses.

### 4.2 Salt sources over time

Figure 5 shows the provenance of the different groundwater types as fraction of the total modelling domain. In all four T-model scenarios, the model domain starts initially mainly fresh, dominated by infiltrated Pleistocene river water. T, his water is then replaced by hypersaline groundwater. As sea-level rises, this hypersaline groundwater is in turn replaced with sea water. The C-M-B-P model scenario takes a slightly different course, as the model is in steady state during stress period 1, dominated by river water. As sea-level rises and the hydraulic gradient decreases, the hypersaline groundwater and infiltrated sea water volumes increase at the expense of river water volumes. Regardless of differences amongst model scenarios, the total volume of sea groundwater is lower than the total volume of hypersaline groundwater and river water in all model scenarios. The total amount of dune water remains small for all five model scenarios over the entire modelled time.

### 4.3 Model evaluation

Figure 6 shows the evaluation results for all palaeohydrogeological reconstructions. Note that the four classes above 5 g TDS $L^{-1}$ in Fig. 1 were grouped into two classes: "saline" (5–35 g TDS $L^{-1}$) and "hypersaline" (35–100 g TDS $L^{-1}$), as the number of TDS measurements was low for these classes. All model scenarios matched the location of fresh groundwater correctly ($\Lambda = 0$) at a majority of the observation points. However, their accuracy worsens with increasing salinity. Despite having sometimes quite different salinity distributions (Fig. 4), the majority of the model scenarios predict the location of the brackish and saline zone with the same error. More striking differences are observed in the hypersaline zone, where we observe a division around $\Lambda = 0.07$ into two groups. There are the scenarios with Md$|\Lambda| < 0.07$, that predict the location of the HGw with a similar skill as the location of saline groundwater. We call these scenarios "acceptable". Specifically,

these are the following five model scenarios: C-M-B-P, C-N-T-P, H-M-T-P, H-F-T-P, H-N-T-P. The other scenarios perform considerably worse in predicting the location of the HGw.

All equivalent steady-state scenarios required at least several thousands of years to reach a steady state (Table 4) from an initial Pleistocene steady state. Most notable are the B-scenarios, where the hypersaline groundwater caused the system to respond very slowly, over tens of thousands of years, thus exceeding the duration of the Holocene. The shortest scenarios were the N-T scenarios, as they did not include HGw and due to the lack of clay layers the salt water did not experience any resistance during its flow upwards from its initial Pleistocene state to the Holocene steady state.

## 4.4 Freshwater volume dynamics and sensitivity analyis

Figure 7a shows that freshwater reserves have declined strongly throughout the Late-Pleistocene and Holocene. In case of the T-model scenarios, first by free convection of hypersaline groundwater in stress period 1, next, for all model scenarios, by sea-level rise (stress period 2) and from 13.5 ka by the marine transgression. The total FGw volumes drop considerably with a factor ranging from 1.9 (C-M-B-P) up to 5.5 (C-N-T-P). There is also considerable variance amongst acceptable model scenario results, with the C-M-B-P having 74% more FGw than C-N-T-P. A peculiar observation for the model scenarios C-N-T-P, H-N-T-P, and H-F-T-P is that during the marine transgression (stress period 3) the total FGw volume recovers after a quick drop. This is caused by the disappearance of the sabkhas, stopping inflow of HGw, whereas outflow of HGw continues, allowing the total FGw volume to recover (see video supplement).

The sensitivity analysis (Fig. 7b) shows a clear influence of horizontal clay layers. For the M-model scenarios, more FGw is maintained in the transient model scenarios than in the steady-state model scenarios. In contrast, the opposite is visible for the N-model scenarios: FGw volumes are lower in the transient simulations than in their steady-state counterparts.

In the model scenarios with marine clay layers there are still considerable fresh groundwater reserves available below 300 m depth and in one case even offshore [C-M-B-P] (Table 5). This table shows that these parts of the model are the most uncertain as well, since disregarding potential deep and offshore fresh groundwater volumes decreases the spread from 74% to 32%.

## 4.5 Fresh-salt distribution: the palaeohydrogeological reconstruction against its equivalent steady-state

$\Delta C$ (eq. 3 in sect. 3.8) is generally negative in the marine clay (M) comparisons (snapshots 2 and 4 in Fig. 8), as the low permeable clays block vertical intrusion during the marine transgression, concurrent with previous research (Kooi et al., 2000; Post et al., 2013). In the no clay (N) comparison (snapshots 1, 3, and 6 in Fig. 8) however, we observe something different: the shallow groundwater (< 200 m depth) is saltier, whereas the toe of the wedge of the steady-state equivalents lays further land inward. This pattern is most clearly visible in the homogeneous model (Fig. 8, plot 1) and is obscured more as heterogeneity increases. In the C-N-T scenario, the palaeohydrogeological reconstruction is more saline nearly everywhere, as there is a larger volume of hypersaline groundwater that is influencing the toe of the seawater wedge (see also Fig. 4, snapshot 3).

In all comparisons, (former) lagoons are visible (Fig. 3 and Fig. 8). In the west, the former Maryut lagoon is clearly visible in all model scenarios as a red zone, stretching all the way towards the Wadi El Natrun depression. The central and eastern lagoons, however, are coloured blue. At these areas, the palaeo-hydraulic gradients were the steepest as the lagoons lay more land inward up till 0.2 ka (Stanley and Warne, 1993), resulting in a quicker freshening of the groundwater at these locations.

5     Figure 9 shows the isosurface separation $\omega$ for the model scenarios where the previous described pattern was visible. An S-shaped trend in $\omega$ is visible with depth, where there is a maximum isosurface separation at around -130 m and minimum at around -350 m depth. The latter is not visible in C-N-T as the hypersaline groundwater influences the fresh-salt interface. This trend means that for these four model scenarios, using a steady-state model results in an underestimation of seawater intrusion length above -250 m depth and overestimation of the intrusion of the "toe" of the wedge. The median of this over-

10    and underestimation of the seawater wedge top and toe ranges up to 10 km. The isosurface seperation increases with concentration, because the denser surface has to rotate further to reach a steady state.

## 5 Discussion

The efforts of conducting 3D palaeohydrogeological reconstructions were rewarded with new insights. We quantitatively show for the first time that a strong reduction in FGw volume presumably occurred in the Nile Delta Aquifer during the last 32 ka. As sea-level rose, FGw volumes were reduced with a factor ranging from 1.9 to 5.4 (Fig. 7). Just as in the Netherlands (Delsman et al., 2014) and the Mekong (Van Pham et al., 2019), the system did not reach a steady state in the last 9 ka. Our equivalent steady-state scenarios required at least 5.5 ka to reach a steady state (Table 4), a period in which already considerable changes occurred to the boundary conditions (Fig. 3). This increased to tens of thousands of years for the more complex models. Using a steady-state approach with current boundary conditions can therefore not result in a reasonable estimate of the current fresh-salt groundwater distribution for such a complex, large-scale system. Moreover, the variance in FGw volumes of the palaeohydrogeological reconstructions was larger than that of the steady-state model scenarios. This implies that conducting sensitivity and uncertainty analyses on steady-state model scenarios, can lead to grave underestimations of the uncertainty. Michael et al. (2016) showed with steady-state models that strong, well connected heterogeneities can lead to chaotic salinity distributions, showing a high spatial variability that we also observe in our most heterogenous cases (Fig. 4). This variability only increases when palaeohydrogeology is accounted for (Fig 7b), because there is less time for mixing to smoothen strong concentration gradients. Furthermore, we have shown that it is physically possible that the influences of the Holocene marine transgression can be still observed in the delta; either as freshwater volumes in between clay layers in the northern part of the NDA, concurrent with earlier research (Kooi et al., 2000), or as a steeper fresh-salt interface than would be expected a priori from an exploratory steady-state model. The latter has to the authors' knowledge never been shown before in this detail. The implications of this steeper interface are that using the steady-state approach can lead to an underestimation of the amount of salt groundwater above roughly 250 m depth and an overestimation of the toe length of the interface (Fig. 9). Likewise, this non-steady, steep interface could explain the observed freshening of the shallow NDA as observed in hydrogeochemical studies (Barrocu and Dahab, 2010; Geriesh et al., 2015), since only the zone above 250 m depth has been sampled. This is the zone that we expect to slowly freshen from our comparison between palaeohydrogeological and steady-state models (Fig. 9). It was however difficult to compare the observed freshening with model results directly, since the time scale over which this observed freshening has developed is often unknown (Stuyfzand, 2008). We can affirm though that it is physically possible that the NDA was predominantly freshened with surface water (Fig. 5), as hypothesized earlier by Geirnaert and Laeven (1992). Deltas with abundant low permeable clay layers are expected to possess larger quantities of deep fresh groundwater than would be approximated with steady-state scenarios. Examples of such deltas are the Chao Praya Delta, Thailand (Yamanaka et al., 2011), Red River Delta, Vietnam (Larsen et al., 2017), and the Pearl Delta, China (Wang and Jiao, 2012). In deltaic groundwater systems consisting of mainly coarse material, a steady-state approximation may underestimate the slope of the fresh-salt interface. Examples of these deltaic areas are the Niger Delta, Nigeria (Summerhayes et al., 1978), and the Tokar Delta, Sudan (Elkrail and Obied, 2013). Note that the Holocene transgression in the above-mentioned Asian deltas reached more land inward than

it did in the Nile Delta (Sinsakul, 2000; Tanabe et al., 2006; Zong et al., 2009), thus their salt groundwater distributions are expected to strongly deviate even more from steady-state approximations than in the research case in this study.

A wide range of model scenarios resulted in acceptable results. This is mainly attributed to the limited availability of salinity observations in the saline zone, which is the area where the results of the acceptable scenarios differed the most. Although the number of observations may look adequate on a 2D map, it is still by far insufficient in 3D. Similar conclusions were drawn by Sanford and Pope (2010) for the Eastern Shore of Virginia (USA), an area with a similar observation density. Though it is generally assumed for the sake of simplicity that the northern part of the delta is completely saline, here we show that there might also be large, overlooked quantities of offshore fresh groundwater. Our model-based FGw volume estimates are on the large side. For instance, Mabrouk et al. (2018) estimated a total amount of 1290 km$^3$ and Sefelnasr and Sherif (2014) 883 km$^3$. We stress that the latter authors disregarded the 80 km of coastal shelf offshore and used a lower porosity value. Reducing the effective porosity from our 25% to their 17%, reduces our estimated FGw volume with 32%; disregarding the increased groundwater flow velocities this would result in, which in turn would slightly increase the extent of the FGw zone.

Of all groundwater constituents, hypersaline groundwater in the NDA is the main contributor to total salt mass (Fig. 5). Apparent from the model evaluation is that the only acceptable model where HGw originates from the bottom includes a closed off continental slope and extending, low-permeable horizontal clay layers, resulting in a compartmentalization of the bottom of the aquifer. When HGw originates from the top, a closed off bottom of the coastal slope is also necessary to preserve the observed amount of hypersaline groundwater since the last Glacial. This implies that regardless of HGw provenance, the interaction at the aquifer-sea connection needs to be very limited at the lower half of the aquifer to explain the observed salinities, as otherwise all HGw would have flowed out to the sea under the steep Late-Pleistocene (32 – 13.5 ka) hydraulic gradients. We can therefore limit the amount of potential combinations between lithology and HGw provenance that were posed in van Engelen et al (2018), as the current study also incorporated a steep glacial hydraulic gradient, whereas van Engelen et al (2018) kept a constant low (interglacial) hydraulic gradient.

Next to estimating FGw volumes, the three-dimensional nature of our model showed us a vulnerable zone in the area: The Wadi El Natrun depression. This area might have started attracting, and eventually draining, sea water from the Maryut lagoon as sea-level rose above the depression height. The amount of evidence of this actually occurring is not overwhelming, but there are some observations in support of this hypothesis (Ibrahim Hussein et al., 2017). Nevertheless, it is concerning that the area with the most far reaching seawater intrusion in our model scenarios, is coincidentally in reality also the area with the most intense groundwater pumping. Even though the 30 years of groundwater extraction in our model scenarios did not seem to show a large influence on seawater intrusion, this influence can increase in the coming century (Mabrouk et al., 2018), since the Nile Delta Aquifer is a slow responding groundwater system. In addition, the large cell sizes of our model can also be the cause of the lack of visible quick responses to groundwater pumping, as large model cells tend to negate local saltwater upconing effects around wells (Pauw et al., 2015). It should also be kept in mind that for future hydrogeochemical

analyses of the Wadi El Natrun area the former Maryut lagoon used to extend a lot more land inward towards this depression than currently, and that the composition of the water in this lagoon fluctuated strongly over time (Flaux et al., 2013).

Despite our efforts at increasing the realism of our model scenarios, some processes were only incorporated in a very simplistic manner. The most prominently simplified process in this arid area is salinization due to evapoconcentration and redissolution of precipitated salts. Though this does not currently seem to influence groundwater salinities, it is hypothesized to have been of influence in the past (Diab et al., 1997; Geirnaert and Laeven, 1992). In our T-scenarios, the Pleistocene sabkhas were assigned a fixed concentration (120 g TDS $L^{-1}$), meaning that free convection here was unconstrained by available salt and water mass. Our T-scenario results indeed show a very rapid change in the first 5 ka (Fig. 5 and 7), after which they reach equilibrium conditions until the next stress period. This may seem very rapid, but experiments with smaller scale models which fully accounted for salt precipitation and dissolution, evapoconcentration, variable-density groundwater flow and the unsaturated zone, showed that higher salinities ( > 200 g TDS $L^{-1}$) are possible under evaporation rates similar to those in the Nile Delta (Shen et al., 2018; Zhang et al., 2015). Given that the Late-Pleistocene sabkhas did not migrate (Stanley and Warne, 1993), we deem our T-model scenarios to be conservative. Likewise, rock salt dissolution is thought to influence salinities around the borders of the delta (Ibrahim Hussein et al., 2017). Either of these processes, evapoconcentration or rock salt dissolution, can explain the saline observations in the South-East (Fig. 1), which our model cannot.

In addition, for a proper physical representation of free convection, a finer grid is required. A coarse horizontal cell size results in a delay in the onset of free convection, while a coarse vertical cell size results in an onset of free convection even for situations that are expected to be stable (Kooi et al., 2000). van Engelen et al. (2018, Appendix D) investigated the errors caused by coarse model cells for the Nile Delta and found that especially the crude horizontal grid size had an influence. They found that this resulted in similar downward fluxes, but a delay in the onset of free convection. This effect, however, was negligible after ~50 years and thus dwarfed by the timescale used for our stress periods. The coarse vertical grid size was not an issue, since the marine transgression occurred over sand with a high hydraulic conductivity, meaning there is a very instable situation and free convection has to occur. We thus think that the errors made in modelling free convection will not impact our conclusions.

Future research for this area would greatly benefit from an enhanced geological model, especially of the lithology of the delta shelf and slope. A lot of measurements were conducted for the petroleum industry (Sestini, 1989), but most of this data is still inaccessible for external researchers. Furthermore, the available publications on offshore geology are written mainly for the petroleum industry, focusing on the area >2.5 km depth, rendering them unsuitable for hydrogeological research of the Nile Delta Aquifer. Therefore, we think research for this area would benefit from a re-analysis of the available data with a hydrogeological perspective. In addition, a larger salinity dataset, especially in the saline zone would help validating groundwater models that are used more frequently in management decisions, e.g. Nofal et al. (2016).

**6 Conclusions**

A 3D variable-density groundwater flow and coupled salt transport model was used for a palaeohydrogeological reconstruction of the salinity distribution of the Nile Delta Aquifer. It was found that large timescales are involved, as steady-state model scenarios required at least 5.5 ka to reach equilibrium. Hence, none of the evaluated
paleohydrogeological scenarios reached a steady state over the last 9 ka, meaning that the transient boundary conditions definitely had an influence on current groundwater salinity. Given the large range variation in delta-architectures analyses, we can conclude that steady-state models are not likely to result in realistic FGw distributions in deltaic areas. Our results also show that the occurrence of past marine transgressions constitute a valid hypothesis explaining the occurrence of the extensive saline zone land inward. Nevertheless, the estimated FGw volumes were subject to considerable uncertainty, due
to the lack of constraining data in this saline zone, mainly due to uncertainty in the permeability of the horizontal clay layers. Regardless, the palaeohydrogeological reconstruction provided several insights. Firstly, during the Late-Pleistocene the total FGw volumes declined strongly, with estimates ranging from a factor 1.9 to 5.4. Secondly, ignoring past boundary conditions lead to an underestimation of FGw uncertainty resulting from a lack of knowledge on geological schematizations and HGw provenances. Thirdly, the differences between the fresh-salt distribution of the palaeohydrogeological
reconstruction and the steady-state scenarios varied, depending on the geology. In case of low permeable clay layers, the palaeohydrogeological reconstruction resulted in considerable volumes of fresh groundwater stored underneath these clay layers. However, with more permeable or no clay layers, the fresh-salt interface was steeper than in the steady-state equivalents. Therefore, using steady-state boundary conditions is likely to result in erroneous fresh-salt distributions and an underestimation of the uncertainty in FGw volumes.

**Video supplement**

Supplementary videos show the development of the groundwater fresh-salinity distribution through time for the five acceptable model scenarios. These can be found under the following DOI: 10.5281/zenodo.2628427

**Appendix A: Additional figures methods**

-Figure A1 here-
-Figure A2 here-
-Figure A3 here-

## Appendix B: Sensitivity analysis of the resistance of the boundaries

To assess the effects of our assumption of the boundary resistance, we ran two alternative versions of the "H-N-T-P" scenario with a different resistance (Fig. B1). This scenario was chosen as it presumably is the "acceptable" scenario that would be affected the most by the resistance value, as it has no horizontal clay layers that resist changes in boundary conditions and the sea boundary has the most open connection with the sea. We multiplied the resistance with a factor 0.5 and 2. Lowering the resistance more than with a factor 0.5 lead to numerical convergence issues. Fig. B1 shows that throughout the Pleistocene the resistance influences the groundwater types, as the lower resistance allows more river water to be replaced with hypersaline groundwater. The groundwater types of the different models quickly converge, however, through the Holocene. We therefore think that the choice of boundary resistance has limited effect on our results and conclusions, despite that we only varied the resistance to a limited extent in this sensitivity analysis.

## Code and data availability

Modelinput files can be found under the following DOI: 10.5281/zenodo.3461667. Scripts and software to reproduce the figures are available in this repository: https://github.com/JoerivanEngelen/Nile_Delta_post with the DOI: 10.5281/zenodo.3461788.

## Author contributions

Joeri van Engelen performed the conceptualization, pre- and post-processing, analysis, and ran the simulations. Jude King interpreted the borelogs and created lithological models. Jarno Verkaik coded the parallel version of iMOD-SEAWAT and helped running the simulations on Cartesius. Eman Nofal provided the data and local expertise. Marc Bierkens and Gualbert Oude Essink supervised this research and helped with its conceptualization and methodology. Joeri van Engelen prepared the manuscript with contributions from all authors.

## Competing interest

The authors declare that they have no conflict of interest.

## Acknowledgements

The authors declare that they have no conflict of interest. Firstly, we would like to thank Benjamin Pennington for kindly providing the palaeogeographical data and answering our questions on palaeogeography. Secondly, we would like to thank Leanne Morgan and one anonymous reviewer for their comments, which significantly enhanced the quality of this paper.

Thirdly, we would like to thank Gijs Janssen for his support in developing the iMOD-SEAWAT code. Fourthly, we would like to thank Huite Bootsma and Martijn Visser for their help with handling model output. Finally, we would like to thank Edwin Sutanudjaja for his tips on running jobs on Cartesius. This work is part of the research programme The New Delta, which is financed by the Netherlands Organisation for Scientific Research (NWO). Additionally, this work is partly financed by NWO and the Ministry of Infrastructure and Water Management under TTW Perspectives Program Water Nexus. This work was carried out on the Dutch national e-infrastructure with the support of SURF Cooperative, on a NWO Pilot Project Grant.

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

**Table 1: Letters that are used to create model scenario codes.**

| Feature | Scenario member | Letter |
|---|---|---|
| Sea connection {*sea*} | Open | O |
| | Half-Open | H |
| | Closed | C |
| Horizontal clay layer {*clayer*} | No clay | N |
| | Fluvial clay | F |
| | Marine clay | M |
| HGw provenance {*prov*} | Top | T |
| | Bot | B |
| Temporal change {*temp*} | Palaeo | P |
| | Steady | S |

**Table 2: References to data used for spatially- and/or time varying input to the model.**

| Data | Temporal | Spatially | Reference | Remarks |
|---|---|---|---|---|
| Late Pleistocene sea-level (Eustatic) | yes | no | Spratt and Lisiecki (2016) | |
| Holocene sea-level (Mediterranean Sea) | yes | no | Sivan et al. (2001) | No detailed sea-levels for the Holocene exist for the Nile Delta, the closest data is from the Levant (Pennington, 2017) |
| Palaeosalinities lagoons | yes | no | Flaux et al. (2013) | Data for Maryut lagoon, taken as representative for other lagoons |
| Palaeogeography | yes | yes | Pennington et al. (2017); Stanley and Warne (1993) | Palaeogeography determines the spatial-temporal occurrence of the sea, dunes, rivers, and lagoons |
| Bathymetry | no | yes | GEBCO (2014) | |
| DEM | no | yes | NASA (2014) | |
| Groundwater extraction wells | yes | yes | Nofal et al. (2018) | |
| Geological borelogs | no | yes | Nofal et al. (2016) | |

**Table 3: Parameters fixed throughout simulation.**

| Parameter | Value | Unit | Reference | Remarks |
|---|---|---|---|---|
| Effective porosity | 0.25 | - | Mabrouk et al. (2018) | |
| Longitudinal dispersivity | 2 | m | Oude Essink et al. (2010) | Value for Rhine Meuse Delta, a similar system |
| Transverse dispersivity | 0.2 | m | Gelhar et al. (1992) | Reference used for anisotropy between transverse and longitudinal dispersivity |
| Vertical dispersivity | 0.02 | m | Gelhar et al. (1992) | Reference used for anisotropy between vertical and longitudinal dispersivity |
| Molecular diffusion coefficient | 8.64E-5 | $m^2 d^{-1}$ | - | |
| Horizontal hydraulic conductivity Holocene fluvial clay | 0.2 | $m d^{-1}$ | Gallichand et al. (1992) | Average value taken here |
| Horizontal hydraulic conductivity Quaternary sands | 75 | $m d^{-1}$ | Barrocu and Dahab (2010); Laeven (1991); Mabrouk et al. (2013) | |
| Horizontal hydraulic conductivity Quaternary marine clays | 1e-4 | $m d^{-1}$ | Domenico and Schwartz (1990) | Upper limit taken here, as these values stem from lab scale measurements and we model on a large scale. |
| Horizontal hydraulic conductivity Quaternary clayey sands | 75 | $m d^{-1}$ | Farid (1980) | Author states that kh clayey sands is close to kh of sands |
| Anisotropy (horizontal divided by the vertical hydraulic conductivity) | 10 (100 for Holo clay) | - | Farid (1980) | |
| Resistance surface water beds | 100 | d | De Lange et al. (2014); Timmerman and Hemker (1993) | Value for Rhine Meuse Delta, a similar system |
| River stage apex delta | 15 | m | NASA (2014) | |
| Salinity sea | 35 | g TDS $L^{-1}$ | - | |
| Salinity sabkha | 120 | g TDS $L^{-1}$ | Van Engelen et al. (2018) | Based on previous model in reference |
| Salinity inflow from hydrogeological base | 120 | g TDS $L^{-1}$ | Van Engelen et al. (2018) | Based on previous model in reference |
| Flux from hydrogeological base | 3E-6 | $m d^{-1}$ | Van Engelen et al. (2018) | Based on previous model in reference |
| Recharge in dunes | 5.5e-4 | $m d^{-1}$ | WMO (2006) | Precipitation near Alexandria |

**Table 4: Time to reach a steady state for all equivalent steady-state scenarios.**

| Code | Time (ka) |
|------|-----------|
| C-F-B-S | 35.0 |
| C-M-B-S | 90.5 |
| C-N-B-S | 23.5 |
| C-N-T-S | 22.5 |
| O-N-T-S | 5.5 |
| O-F-T-S | 10.5 |
| O-M-T-S | 10.0 |
| H-F-B-S | 37.5 |
| H-F-T-S | 7.5 |
| H-M-B-S | 44.5 |
| H-M-T-S | 10.0 |
| H-N-B-S | 19.5 |
| H-N-T-S | 5.5 |

**Table 5: End state FGw volumes for acceptable model scenarios.**

| Code | Total fresh groundwater ($km^3$) | Fresh groundwater < 300 m ($km^3$) | Fresh groundwater < 300 m and onshore ($km^3$) |
|------|------|------|------|
| H-M-T-P | 1974 | 1853 | 1829 |
| C-M-B-P | 2659 | 2325 | 1989 |
| C-N-T-P | 1526 | 1512 | 1511 |
| H-N-T-P | 1728 | 1642 | 1641 |
| H-F-T-P | 1765 | 1670 | 1668 |

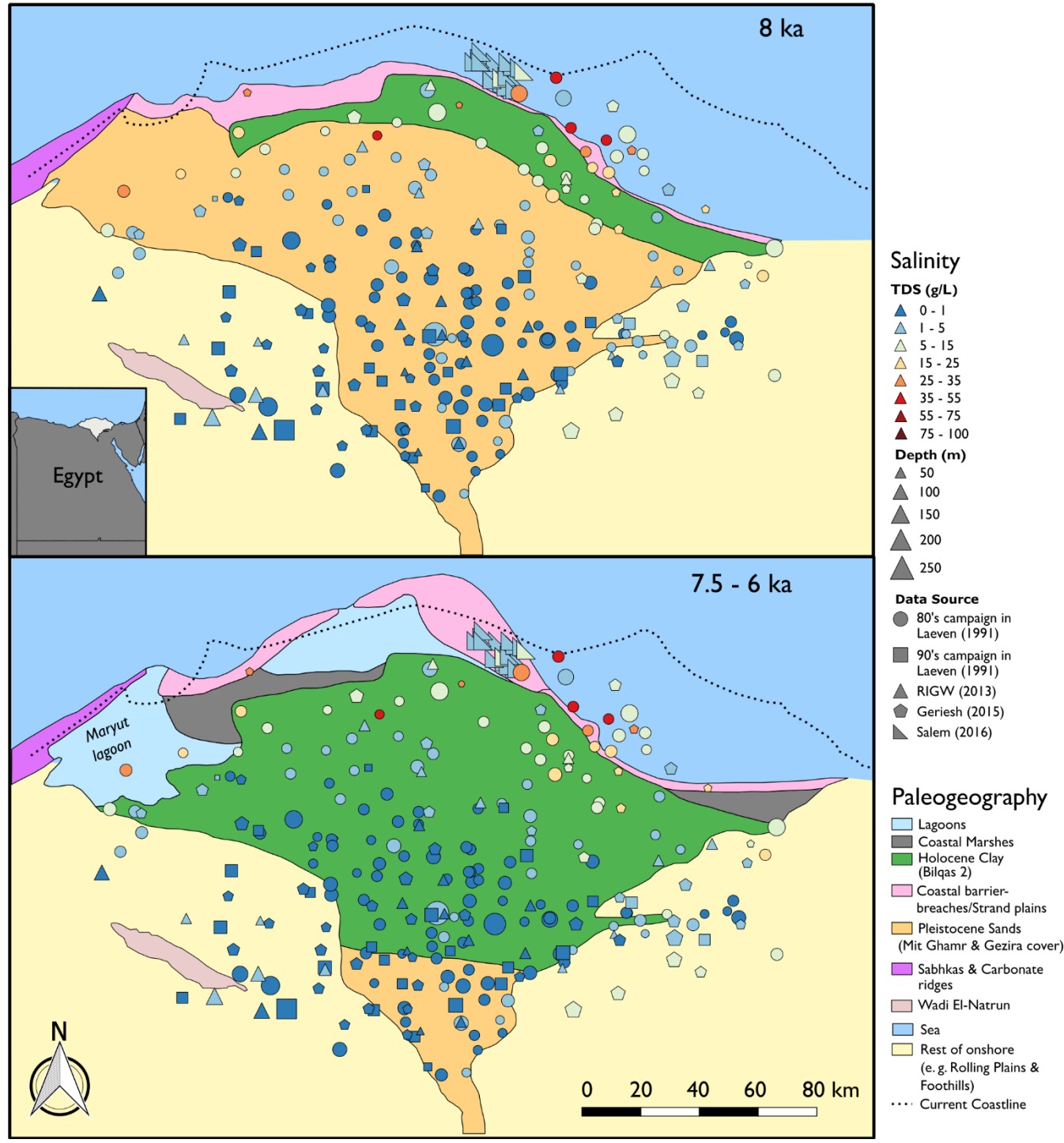

**Figure 1: Palaeogeography and groundwater salinity measurements up to 250 m depth of the Nile Delta. Palaeogeographical data from Pennington et al. (2017). The inset shows the location of the delta in Egypt.**

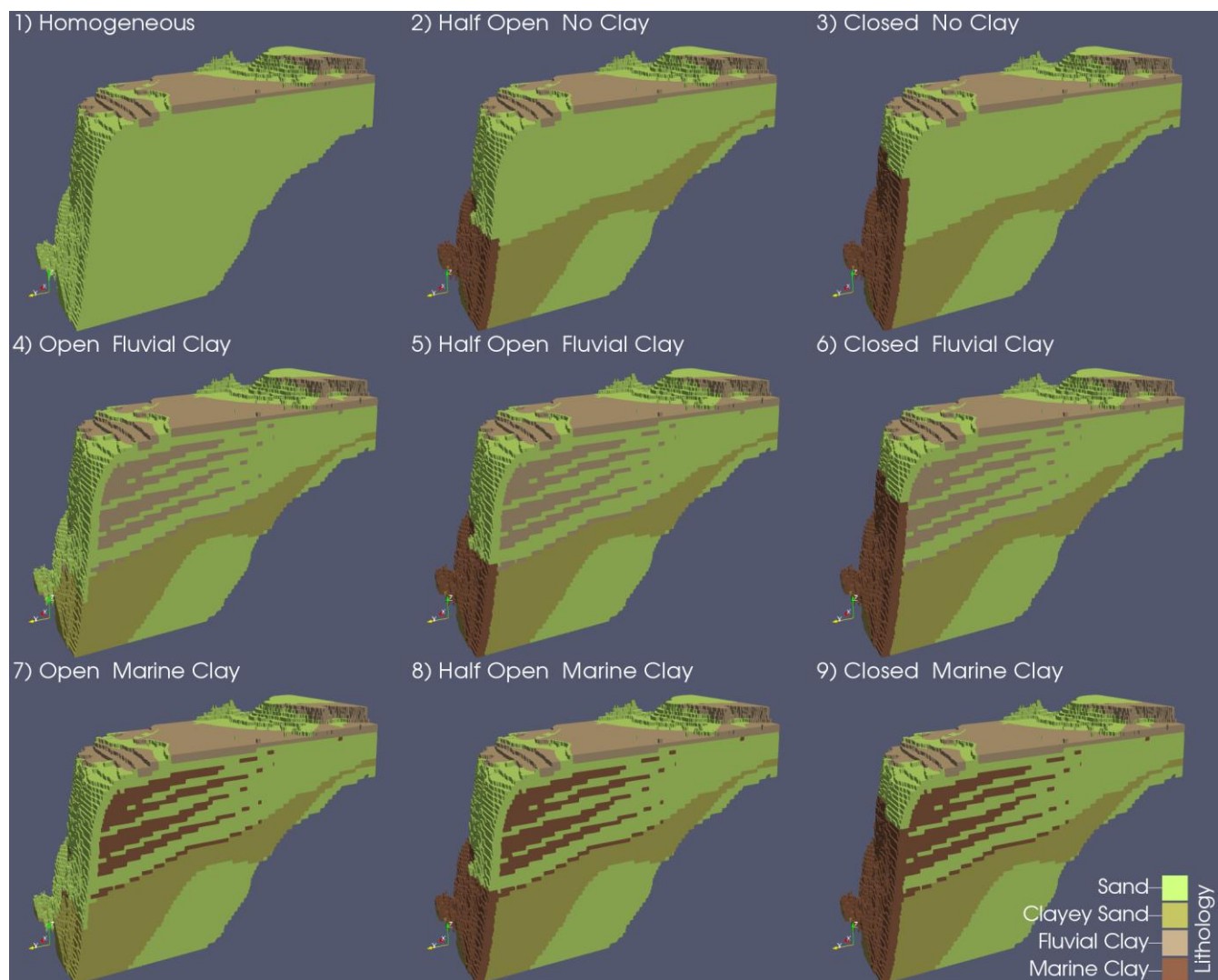

**Figure 2: Snapshots of the lithological model scenarios that are used as input for the numerical variable-density groundwater flow model. The eastern half of the model is plotted here (See Fig. A1 for extent). The first word refers to the connection to the sea of the deeper part of the NDA, the second word to the assigned hydraulic conductivity of the onshore-reaching clay layers.**

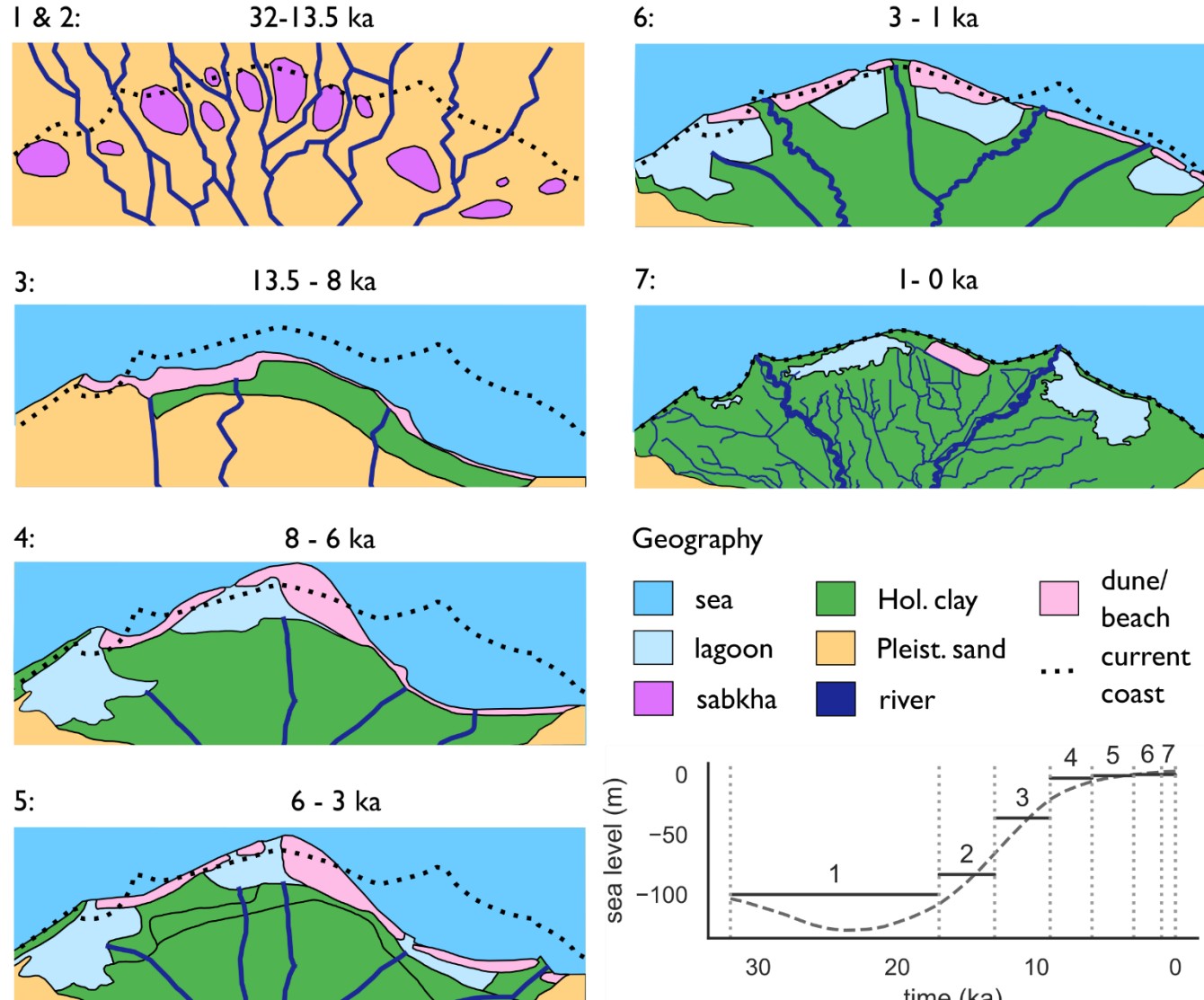

**Figure 3: Stress periods with simplified palaeogeography used as boundary conditions. Figures modified after (Pennington et al., 2017; Stanley and Warne, 1993). The panels focus on the area around the present-day coastline (see Fig. A1 for extent), where upper boundary conditions varied the most. Actual model domain extended beyond the extents of these panels. In the bottom right: The median eustatic sea-level curve (Spratt and Lisiecki, 2016) and the selected sea-level for each stress period.**

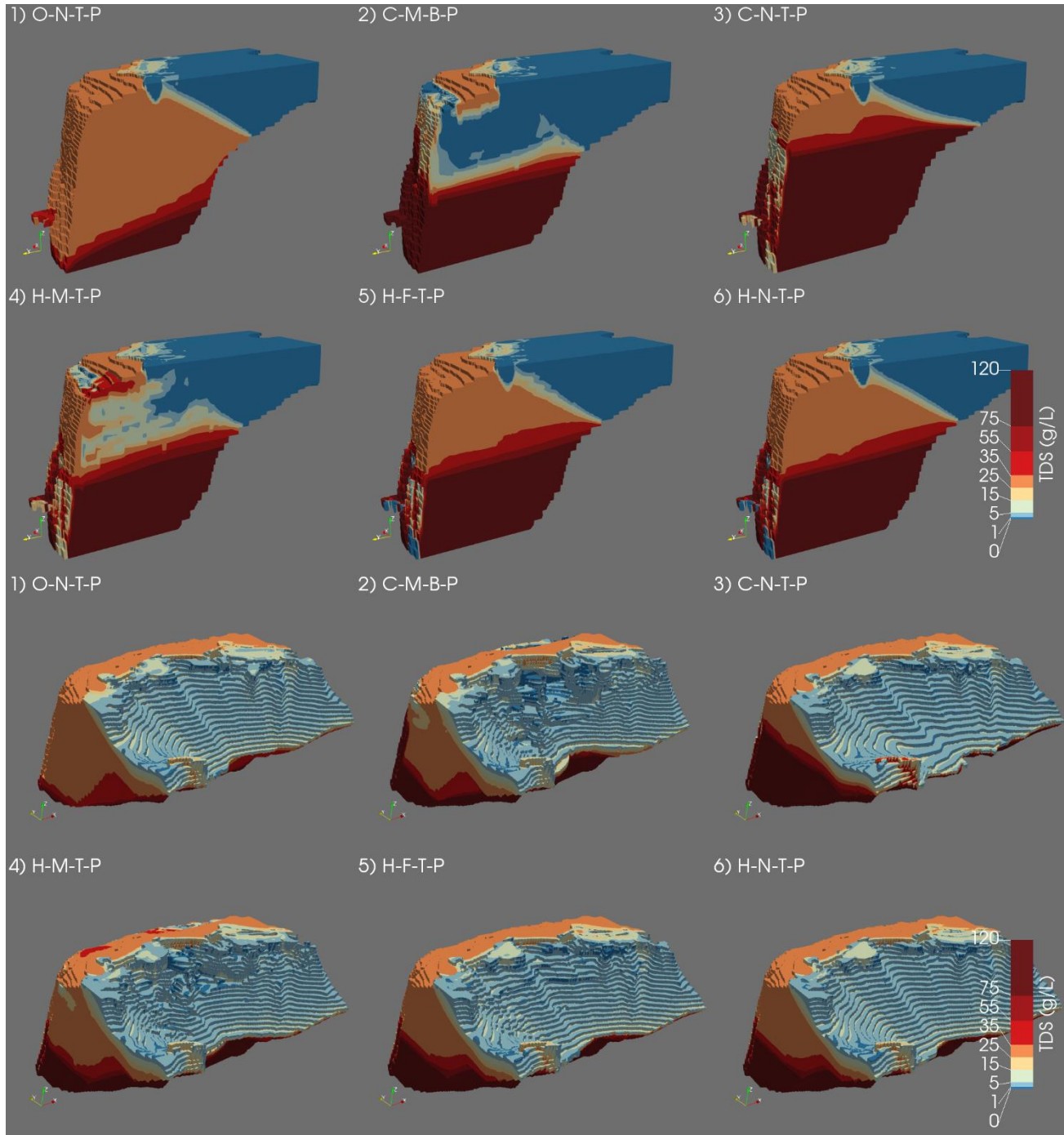

**Figure 4: Salinity distributions of the five selected model scenarios after the validation; the results of the homogeneous model (viz. O-N-T-P) are also added as first plot for reference. Note that the results are stretched in the z-direction by a factor 150. In the top six plots the salinity distribution is sliced in half over the model domain and camera is pointed in the south-east direction. In the bottom six plots the full delta is visible, but the fresh groundwater is made fully transparent. Camera is pointed in downwards, in north-easterly direction.**

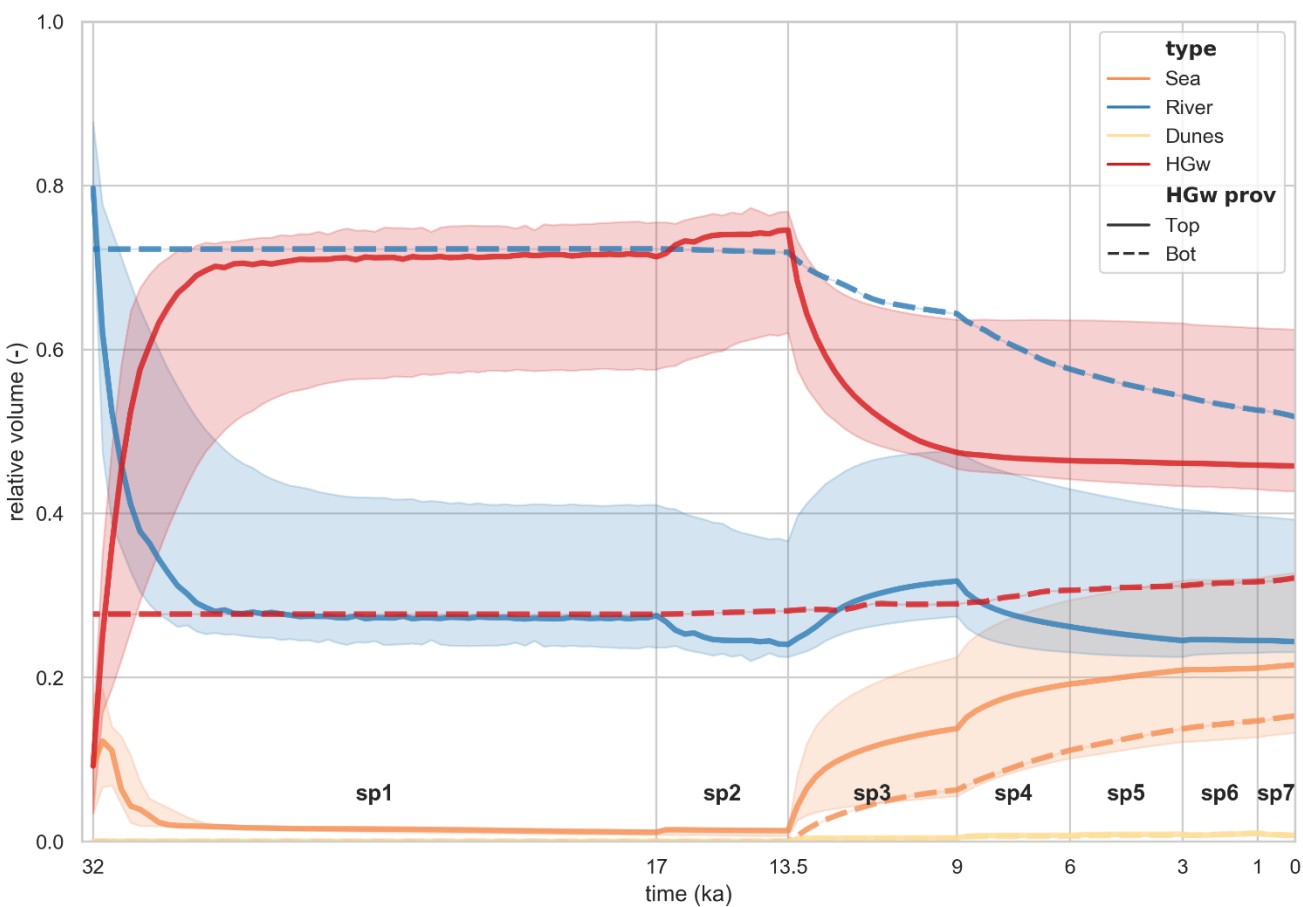

**Figure 5: Groundwater provenance of different water types as fraction of the total modelling domain for the five acceptable model scenarios. The dashed lines represent the C-M-B-P model. Shaded area indicates the range of the acceptable T-model scenarios (C-N-T-P, H-N-T-P, H-F-T-P, H-M-T-P), the thick line their median. Stress period numbers are indicated in bold, preceded by "sp".**

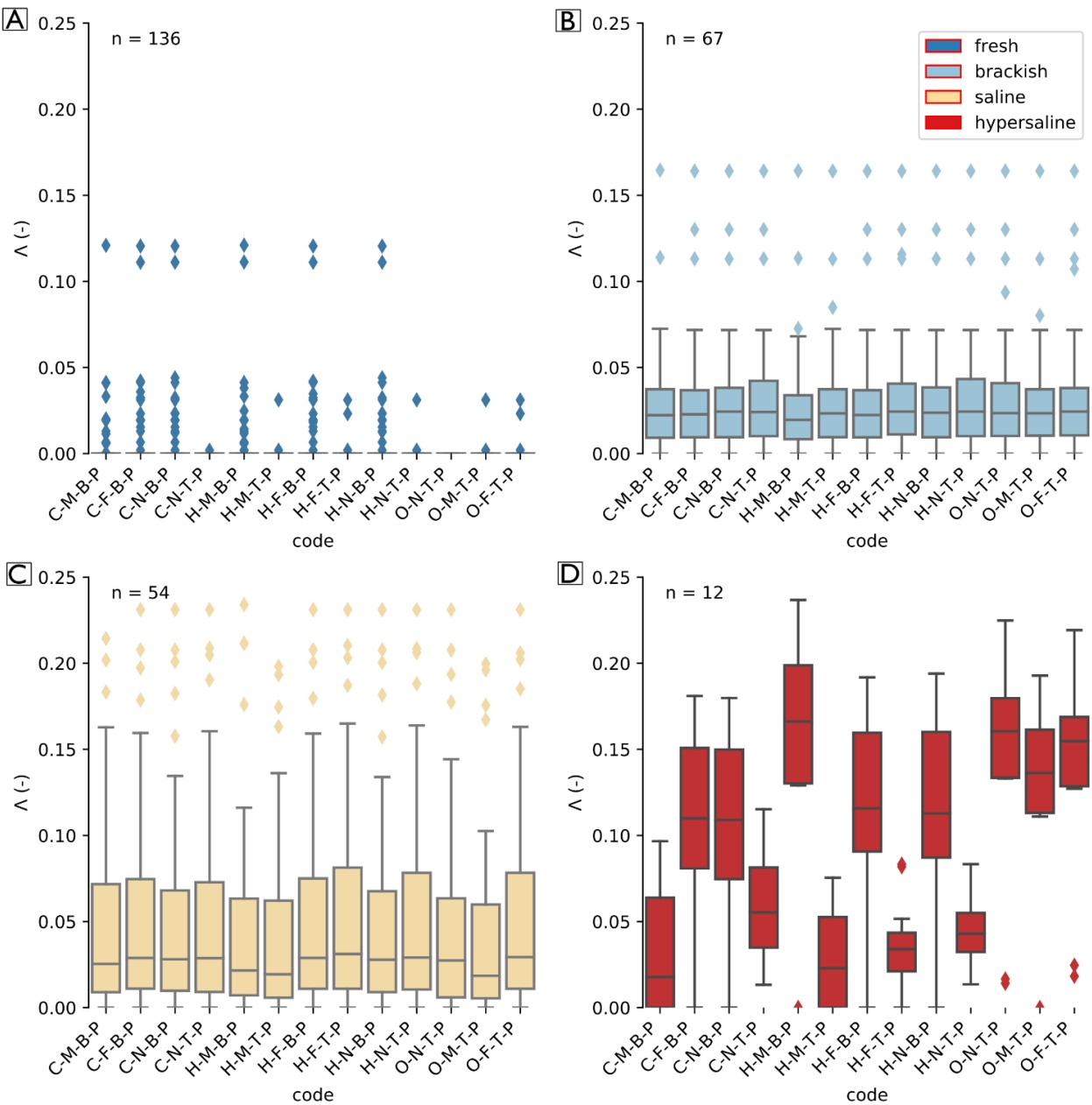

**Figure 6: Goodness of fit boxplots for all palaeohydrogeological reconstructions, binned in to four salinity classes. The higher the value of Λ, the worse the fit. Codes indicate model scenarios (Table 1). TDS values are binned in the classes [0, 1], [1, 5], [5, 35], [35, 100] g TDS L$^{-1}$ for respectively "fresh", "brackish", "saline", "hypersaline", these are respectively plotted in panels A, B, C, D. "n" indicates the amount of observations available in the TDS bin to evaluate model results with. Diamonds indicate outliers, defined as values separated from the first or third quartile at 1.5 times the interquartile range. When no box is plotted for a scenario, 75% of the measurements equal zero, which consequently causes the interquartile range to be zero, rendering every non-zero value an outlier.**

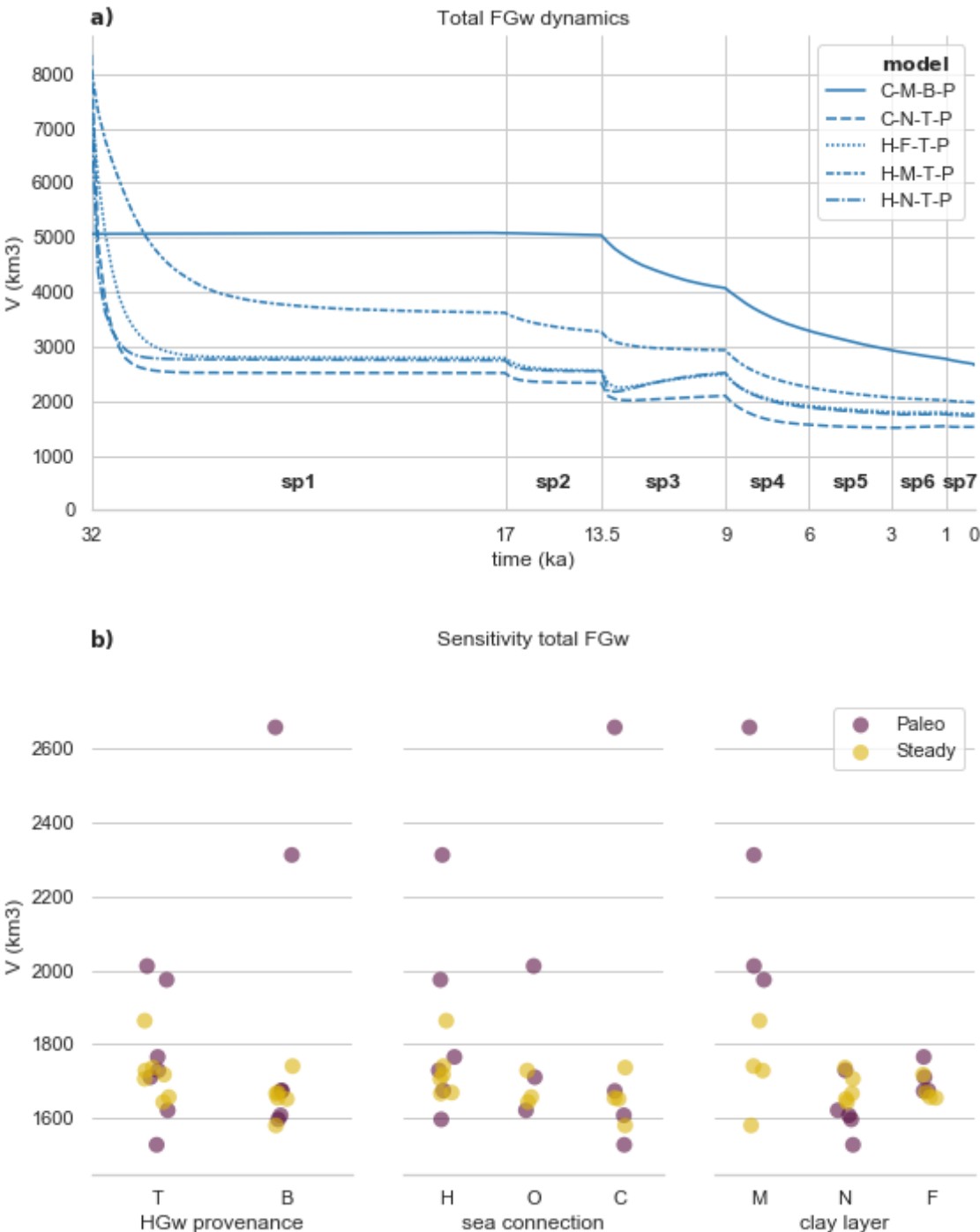

**Figure 7:** (a) the development of the total FGw volume through time for the five acceptable model scenarios. Stress period numbers are indicated in bold with "sp". (b) End state total FGw volumes of all model scenarios, grouped for different inputs.

Colours of the dots indicate a steady state or palaeohydrogeological reconstruction result. Dots are jittered to better show overlapping dots.

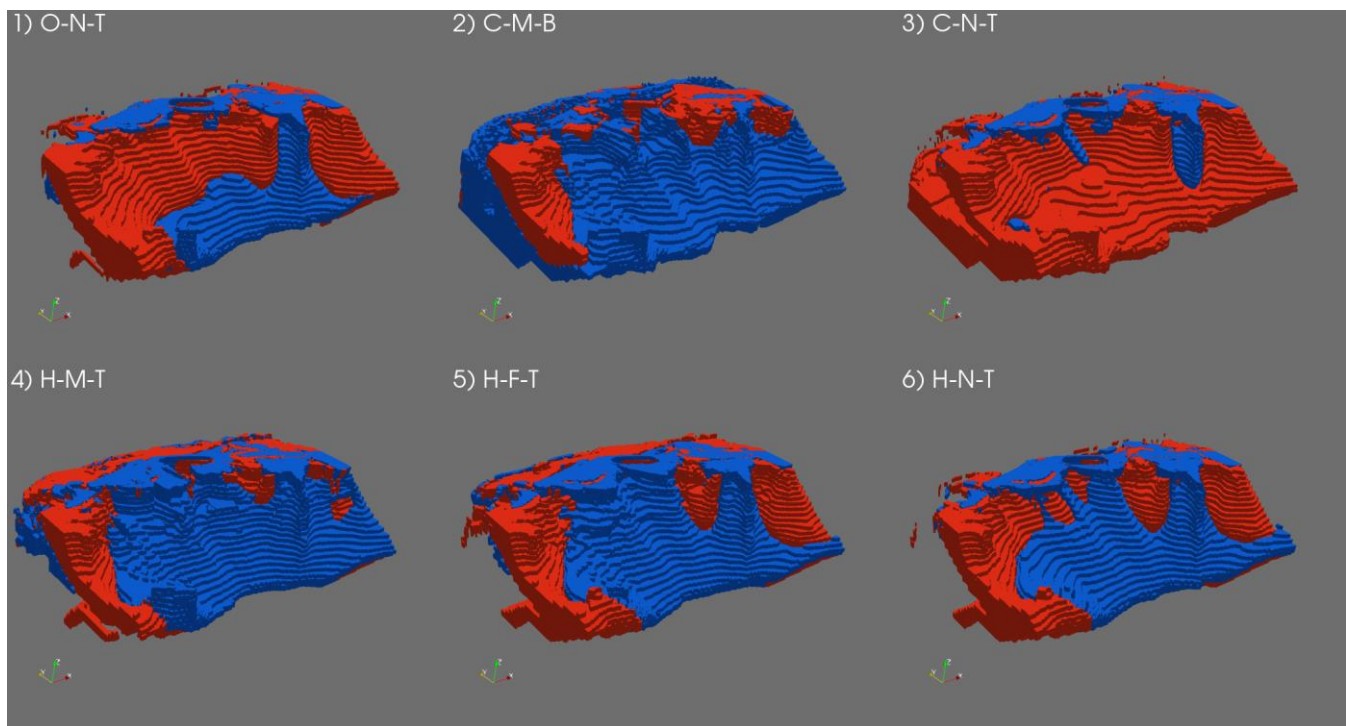

Figure 8: Comparison between the palaeohydrogeological reconstructions and the steady-state model scenarios. Red colours indicate that $\Delta C > 1$, meaning that the palaeohydrogeological reconstruction is saltier than its steady-state equivalent. Vice versa for blue, where $\Delta C < -1$. All values in between -1 and 1 are made transparent to focus purely on major differences. Groundwater below 450 m depth is not plotted, as the present-day fresh-salt interface was not located below these depths. Camera angle is the same as in the six bottom plots of Fig. 4.

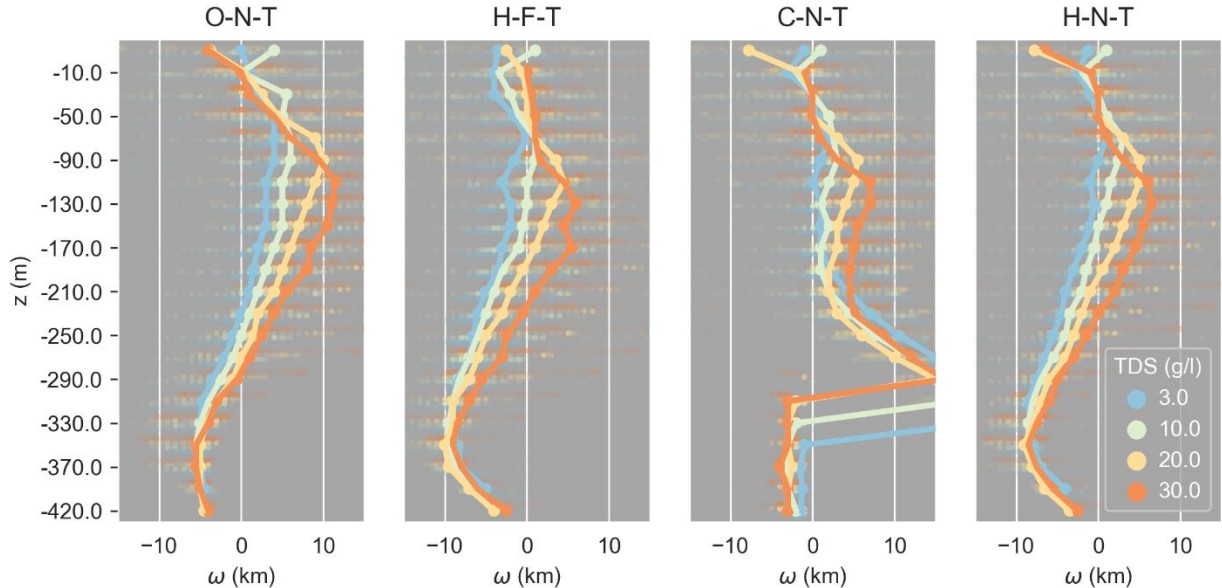

**Figure 9: Isosurface separation ω with depth for different TDS values and selected model scenarios. The line indicates the median isosurface separation, the near transparent dots all data points. Positive ω means the isohaline of the palaeohydrogeological reconstruction is located more land inwards than its' steady-state equivalent and vice versa.**

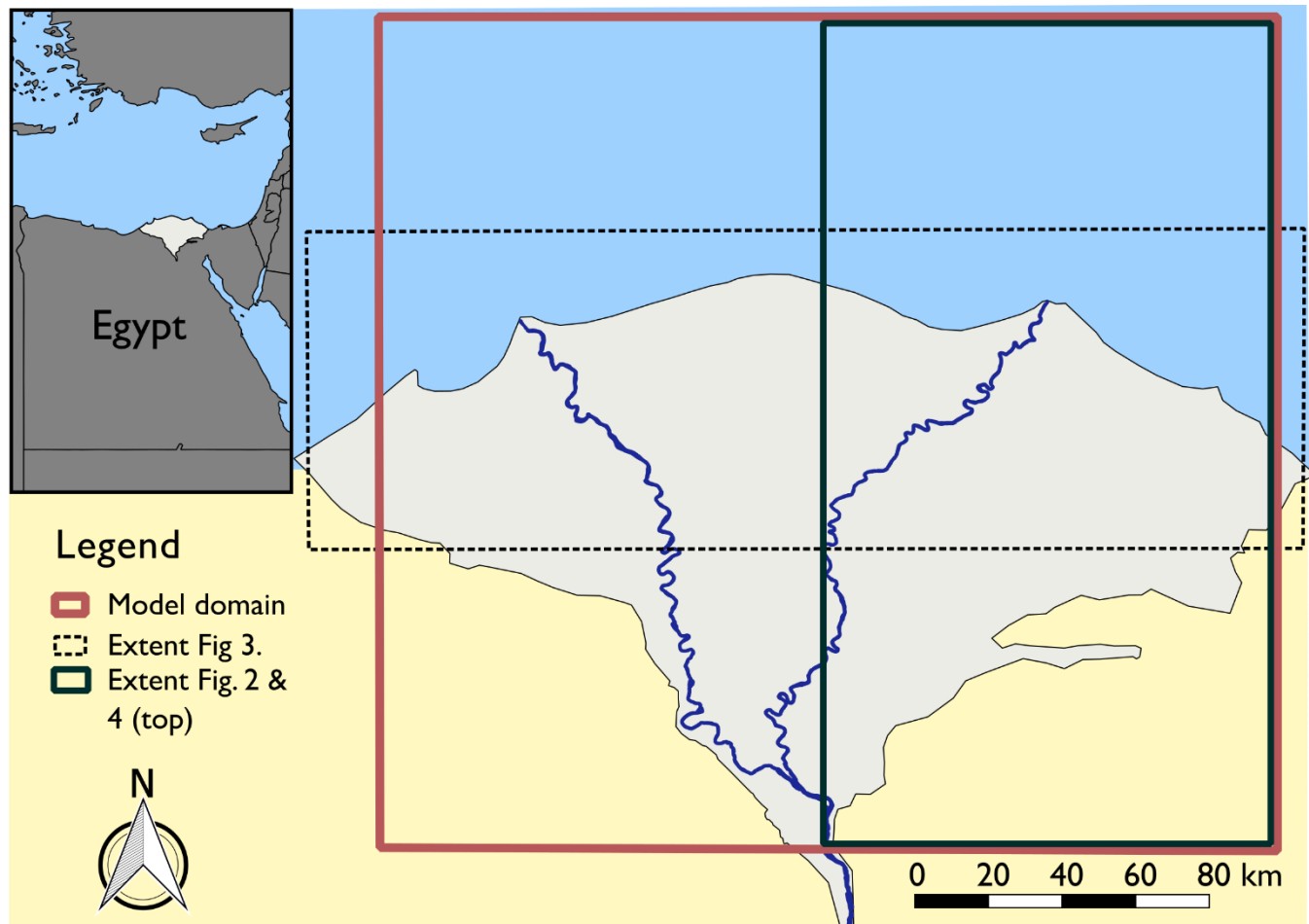

**Figure A1: Overview of the Nile Delta with boxes indicating the model domain and the extents of several figures. The main Nile river branches are plotted as a geographical reference. The red rectangle shows the extent of our model domain, the dark green box the plotted area in Figure 2 and the top half of Figure 4, and the dotted box the plotted area in Figure 3. The inset on the upper left corner shows the location of the Nile Delta in Egypt.**

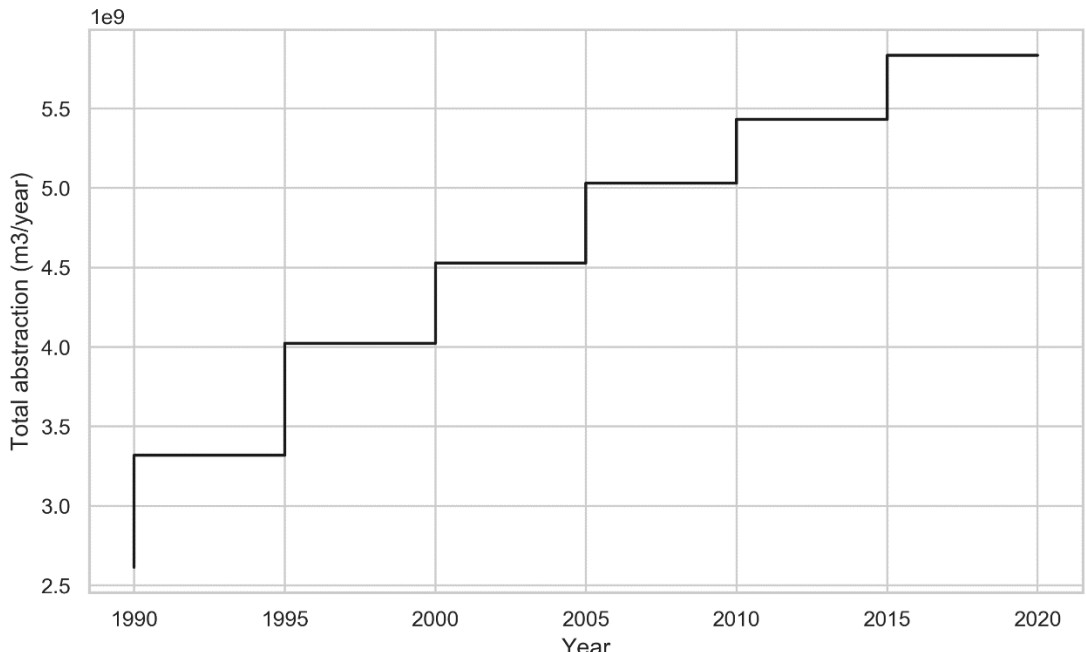

**Figure A2: Total annual abstraction used as model input for the last 30 years of the simulation. This period has stress periods of 5 years.**

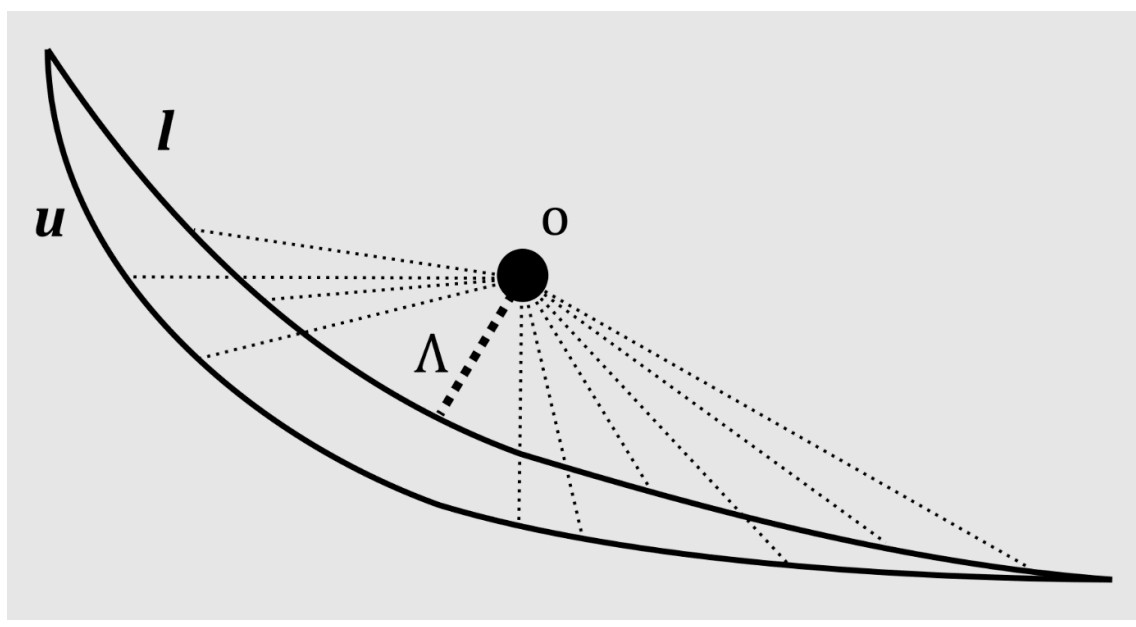

**Figure A3: Sketch of how Λ is determined with equation 1 for a 2-dimensional example. The circle represents the location of the observation.** $l$ **and** $u$**, the isosurfaces of the lower and upper bound, respectively. The thick dotted line represents the minimum Eucledian distance to an isosurface bound, Λ. The thinner dotted lines some other Euclidian distances.**

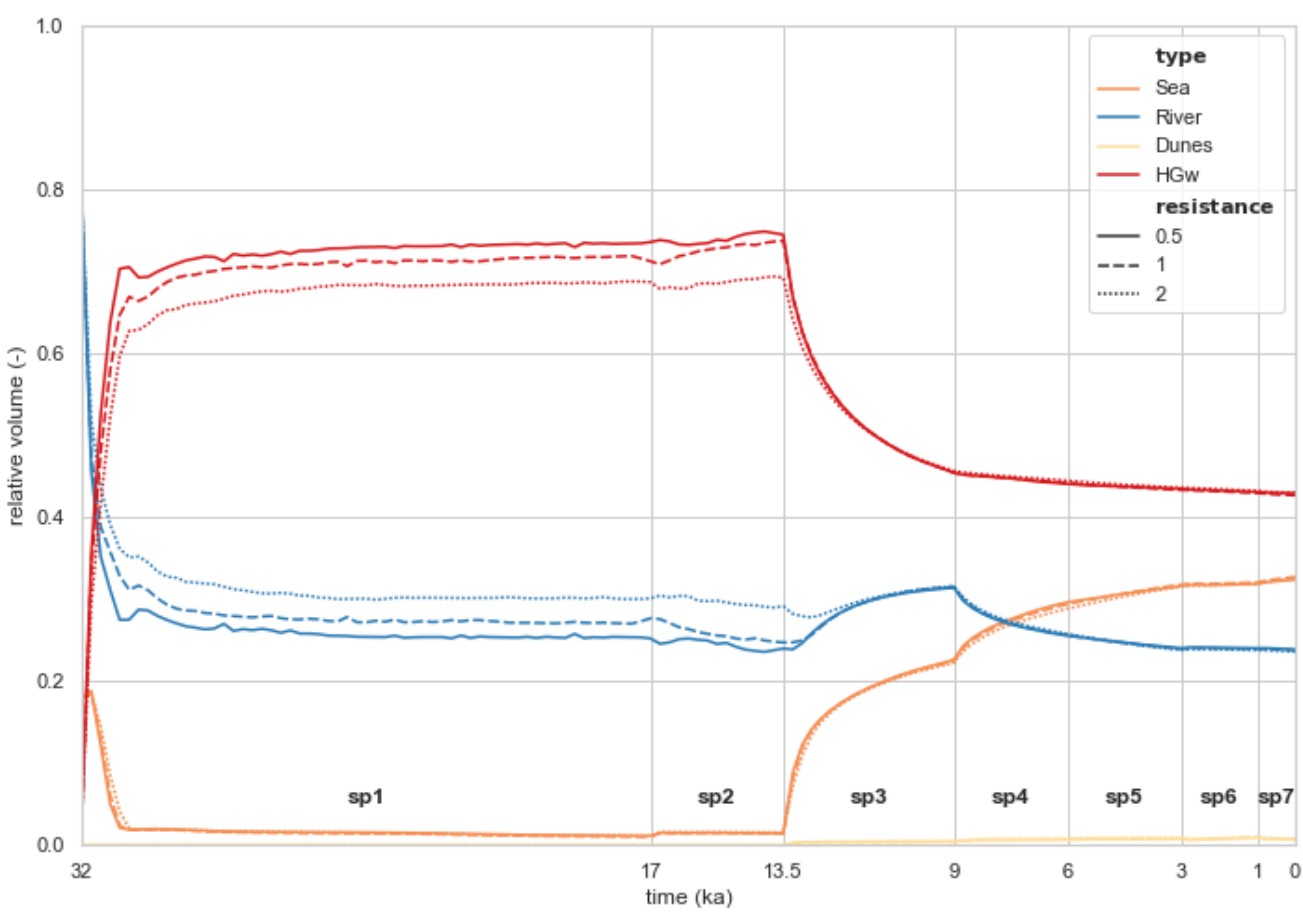

**Figure B1: Groundwater origins for scenario "H-N-T-P" with three resistances. The colour indicates the groundwater type, the linestyle the value the original resistance is multiplied with. Stress period numbers are indicated in bold with "sp".**