# Peer review of "A three-dimensional palaeohydrogeological reconstruction of the groundwater salinity distribution in the Nile Delta Aquifer"

_Hydrology and Earth System Sciences, 2019_

## Referee Comment (RC1) · Anonymous Referee #1 · 3 Jun 2019

This paper addresses the important topic of the groundwater salinity distribution in large-scale (delta) aquifer systems in relation to the palaeo-geographical evolution. The authors use the Nile Delta Aquifer to investigate how different conceptual models affect the simulated present-day groundwater salinity patterns. The results are relevant to comparable systems elsewhere, and the study is rather unique in terms of the large number of model scenarios that was considered. This makes it a valuable contribution, especially because there are few studies of this type. Nevertheless, there is room for considerable improvement, in particular in the way the model scenarios are evaluated and compared.

[Figure]

First of all though, with regards to the Introduction section, you can build a stronger justification for this study by first discussing the studies that have been conducted in other areas (presently starting on page 3, line 25), which demonstrate that over-simplified models of large-scale aquifer systems are conceptually flawed. Then you can bring in the Nile delta system, and argue that there is also a need there to analyze the palaeo-geographical evolution in order to understand the present-day conditions. By starting off with a focus on the Nile straight away, you present it too much as a regional problem, not a scientific analysis that yields outcomes that can be transferred to other study regions.

Moreover, the paper as a whole, but the Discussion section in particular, is a bit of a confusing mix of a number of problems. There is (A) the scientific problem of understanding the evolution of the groundwater salinity distribution in large delta systems over long timescales. This is mixed with (B) the local management problem/question of how much freshwater there is in the Nile delta. And then there is (C) the problem that previous models have assumed steady-state conditions. For a publication in HESS, the local management problem (B) is not the most important. The scientific problem (A) is, and it should be made clear from the onset and throughout the paper that this is the main focus. The implications for the local management problem can be mentioned toward the end (it is especially interesting to note that depending on the conceptualization, the locations where freshwater occurs will differ), but should not feature prominently anywhere else.

With regards to the assumption of the system being in steady-state (C), a considerable portion of the paper is devoted to a comparison between dynamic (i.e., models considering the palaeo-geographical evolution) and steady-state conditions (equivalent simulations for the same geology). It is concluded that the dynamic models are a better fit to the data. But given that the data set has severe limitations (there are relatively few data points, and the quality of the available data is also not assured, at least the paper does not describe the QC/QA procedures), one could wonder if that

is really a such strong criterion. I think a much stronger argument could be made by looking at the time required to reach steady state. This information is not presented for all model scenarios, but in line 28 on page 11 it is mentioned that it took 60 ka for the B-model scenarios. Doesn't this automatically invalidate the steady-state assumption, without having to perform anymore detailed comparisons between the dynamic models and their steady-state equivalents? I am not sure what time is required for the other simulations, but I am guessing it will be on the same order, except maybe for the most unrealistic representations of the delta's lithology. I would encourage the authors to present the timescale aspect in more detail and use it to build the argument against steady-state being a realistic assumption. Much of the detailed comparisons such as those presented in figures 9, 10 and 11 could then be omitted.

This would also have the benefit that the paper becomes more easy to follow, because as it stands, one quickly gets lost in the many different scenarios. The Results section could start with what is now subsection 4.2, which could be expanded and/or merged with subsection 4.4. This would give the reader a much better overview of the actual processes before diving into the more detailed analysis of model performance and freshwater volume.

Page 1 lines 15-16, "observed by hydrogeochemists": No need to suggest a disciplinary bias line 17, "palaeo-reconstruction": This suggests your reconstruction itself is ancient. Choose better wording line 18: Insert "a" between "using" and "state-of-the-art model" line 23: You use both "sea water" and "seawater" in your manuscript. Pick one, and check for consistency. line 29: add "s" to "distribution"

Page 2 line 6: insert "that they were" between "indicated" and "pumping". More generally, the language usage needs some attention, the paper is generally well written, but every now and then it lacks some attention to detail. I will not focus on these issues from this point onward, but the authors should do a careful proofread of their revised manuscript to resolve them.

[Figure]

Page 4 line 31: delete "hypersaline and", the fact that they are hypersaline should not be a reason to discard them. On the next page you talk about hypersaline groundwaters as well (or at least, salinities greater than seawater)

Page 5 line 1: I am not sure if this reasoning holds true. A 1000y old groundwater can still move appreciable distances over a couple of decades if it is near a large well field line 26: Some additional information is required here to explain the choices and rationale behind these 9 different lithological models

Page 6 line 5: Replace "its'" with "its" lines 10, 11: Not sure what you mean by "keeping memory locally". line 15 a.f.: You need to include a map with the model area. lines 23, 24, "as the hills above this height have no important contribution to the groundwater flow": Without the aforementioned map it is hard to assess the validity of this statement. Where are these hills? How high are they? More importantly, what is the recharge and the water table elevation. The elevation of the hill itself is not so important, its hydrogeological characteristics much more so, of course...

Page 7 line 3, "time slices": This sounds like what we would normally call a stress period in groundwater modelling. Why the confusing terminology? And why with a space in the title, and without in this sentence. Please pay more attention to detail. line 12: Replace "announced" with "at" line 22, "100 day resistance" (better would be "a resistance of 100 days" BTW): How sensitive are the modelling results to this assumption? A single value is of course highly unrealistic, and citing studies from the Rhine-Meuse Delta does not provide any justification, because this parameter is just as uncertain and spatially (and even temporally at this timescale) variable there. But you've got to work with what you have, I understand that, but in the end, some sensitivity analysis is required to test to what extent the study outcomes might be affected by this modelling choice. line 31: replace "acquiring" with "achieving"

Page 8 lines 3-6: Did you do this via SEAWAT's density options in the CHD package? A range of 2-18 g TDS/L gives quite a range in density. What value did you adopt? And

again, how sensitive is it?

Page 10 line 7: You could cite the following article here: Sanford, W.E. & Pope, J.P. Hydrogeol J (2010) 18: 73. https://doi.org/10.1007/s10040-009-0513-4 line 10 a.f.: This is somewhat hard to follow and it might be worth adding a sketch that illustrates the principle (could be in a supplementary document). Other authors may choose to adopt this methodology, hence it could be worthwhile doing this.

Page 11 line 19: what do you mean with "behavioural"? And what is the justification for using 0.07? lines 19-21: Sentence does not seem to flow well due to a grammar error, not sure what you are trying to say here.

Page 12 line 3: Not clear what you mean here with "behavioural" line 4: Up until this point the difference between fluvial and marine clay layers has not been explicitly discussed. lines 7, 8: replace nondescript terms like "more 3D patterns are visible in the salinities" and "partly has a conical" with more accurate descriptions lines 24 a.f., "This table also shows...": I could not follow what you are trying to say here

Page 14 line 2 a.f.: I think the point you want to make here is that you can come up with multiple models that fit the observations equally well and yet, the volume of freshwater varies a lot. What do you mean with "The variance in the results should also affect management decisions."? I think the management decisions will not be based on the total volume of freshwater, but on the possibility to be able to extract groundwater in a particular region. See also general remark made before about the relevance of the freshwater volume issue for this paper. I would not start the Discussion section with this paragraph (see next point)

Page 15 line 6: Start the Discussion section with this paragraph, it is much more relevant to a broad readership than local aspects such as the discussion of the total freshwater volume (also see general comment) and flow to Wadi El Natrun.

line 32: Also include a discussion of the representation of free convection phenomena

[Figure]

in your model. The large grid size is prohibitive for an accurate process representation. How confident are you that this does not harm the general conclusions drawn from the model outcomes?

Figure 1: It could just be because of the pdf, but the resolution is very poor. Figure needs an inset showing the location of the area within Egypt/the Mediterranean region. Add north arrow and scale bar! In this figure all areas outside the delta are white, better to make the Mediterranean blue and the desert yellow(-ish).

Figure 2: Somewhere we will need a map with the model boundary. On this map you will need to indicate which part of the model is shown here.

Figures 4, 5: Take the reader by the hand here as the figures are complex. Explicitly mention in the caption that a, b, c and d reflect the salinity classes, and explain the n value.

---

## Referee Comment (RC2) · Anonymous Referee #2 · 13 Jun 2019

General comments: This paper describes numerical modelling used to reconstruct the salinity distribution in the Nile Delta aquifer over the past 32 ka. The model simulations include the Holocene marine transgression and various scenarios relating to possible aquifer lithology, and sources of hypersaline seawater. Through consideration of palaeo-environments, the authors aim to better represent the Nile Delta aquifer and an observed freshening. The authors state that simulation of the freshening of the aquifer has not been possible previously using realistic model parameters and steady-state modelling. The objectives of this modelling study are stated as being (p4, L8) to: 1) investigate the physical plausibility of the Holocene transgression hypothesis for the Nile Delta; 2) investigate the influence of the uncertain geology; 3) provide estimates

of the present-day fresh groundwater [FGW] volumes; 4) assess the importance of using palaeo-reconstructions compared to less expensive steady-state modelling. These objectives have been addressed within the study and the description of how this was achieved is generally clearly detailed. There are not very many palaeo-reconstruction type studies looking at saltwater-freshwater dynamics in coastal aquifers and therefore there is some novelty in this contribution. However the significance is not clear. The study relates specifically to the Nile Delta aquifer and the more generalised extension of the science is not apparent. As such the study is primarily a case study, albeit with some interesting insights for other similar deltaic systems. For this manuscript to be suitable for publication in HESS the authors need to indicate how the study provides a substantial contribution, and is not just a case study.

Specific comments: The use of iMOD-SEAWAT has presumably assisted in reducing run times and making modelling of the various scenarios used in the study tractable. The paper would benefit from more information describing the run times and benefits of using iMOD-SEAWAT. I expect that other researchers will find this interesting for similar palaeo-reconstruction type work.

The study involved numerous scenarios and I found many of these confusing as I read through the paper. The section describing the coding of the scenarios needs improvement. For example, I was not able to discern the meaning of P in the scenario H-F-B-P (p10, L1), despite that this was used as part of the explanation of how the coding system works. Additionally, the term 'behavioural' model scenarios (p. 12, L1) seems odd and I suggest changing.

---

## Author Comment (AC1) · 10 Jul 2019

Dear reviewer 1,

We would like to thank you for your valuable input and insightful comments, the vast majority was implemented in the second version of our manuscript. Please find in the supplement the answer to your reviews itself, a clean second version of the manuscript and also the track changes document (compared to the first version) for easier orientation in the changes made to the manuscript.

Once more thank you very much for your time and input.

[Figure]

Joeri van Engelen, Jarno Verkaik, Jude King, Eman Nofal, Marc Bierkens and Gu Oude Essink

Please also note the supplement to this comment:
https://www.hydrol-earth-syst-sci-discuss.net/hess-2019-151/hess-2019-151-AC1-supplement.zip

---

## Author Comment (AC3) · 10 Jul 2019

Dear editor,

We have answered both reviewers now. Despite that, the system tells us that the manuscript is still at the "waiting for Final Comment" stage. This raises some concern and confusion for me, as I am just about to leave for holiday. I hope we filled out everything correctly.

We have also modified the title of our paper, I hope that also does not raise too much trouble.

[Figure]

For your convenience I have attached both author comments and the paper with- and without markup in a .zip file.

Thank you in advance for your time and input,

Joeri van Engelen

Please also note the supplement to this comment:
https://www.hydrol-earth-syst-sci-discuss.net/hess-2019-151/hess-2019-151-AC3-supplement.zip

---

## Author Comment (AC4) · 10 Jul 2019

My apologies,

For some reason the "waiting for Editor decision" e-mail was sent to my spamfolder (even though the other HESS e-mails were not sent there). I think it is now resolved.

Kind regards, Joeri
* * *

---

## Author Response (AR1)

**Answer to Reviewer 1**

We thank reviewer 1 for his/her thoughtful review that greatly helped to improve our paper. The feedback and suggestions were very motivating for us to improve our paper. In this document, we have entered your comments in italics, added our response as regular text, and subsequently added suggested changes to the paper in red. After our responses you can find the suggested changes made to the text in red. We also like to add that we added supplementary information to this document, as it spares you the effort of looking for an appendix of a previous paper we wrote. Line numbers are written as "**[P… L…]**", indicating page number, line number in the no markup document.

**General comments**

*First of all though, with regards to the Introduction section, you can build a stronger justification*
*for this study by first discussing the studies that have been conducted in other*
*areas (presently starting on page 3, line 25), which demonstrate that over-simplified*
*models of large-scale aquifer systems are conceptually flawed. Then you can bring in*
*the Nile delta system, and argue that there is also a need there to analyze the palaeo-geographical*
*evolution in order to understand the present-day conditions. By starting*
*off with a focus on the Nile straight away, you present it too much as a regional problem,*
*not a scientific analysis that yields outcomes that can be transferred to other study*
*regions.*

This is a good suggestion, as it shifts the focus of the paper to the scientific problem instead of the regional problem. This will also help with Reviewer 2's main concern. We have added sentences of introduction to the broader problem **[P2 L1-7]** and consequently followed your suggestions. Given the severity of the local problems, we subsequently moved the former first paragraph to a dedicated section under "Area description" **[P4 L2-21]** .

The paper now starts as follows:

Palaeohydrogeological conditions have influenced groundwater quality in the majority of large-scale groundwater systems, since groundwater below 250 m depth is dominated by groundwater with an age of over 12 ka (Jasechko et al., 2017). These conditions can especially be found in deltaic areas, where the effects of marine transgressions are often still observed in groundwater salinities (Larsen et al., 2017). More specifically, their low elevation allowed for far reaching marine transgressions, leading to a large vertical influx of sea water, and hampered subsequent flushing with fresh water after the marine regression. This hypothesis is supported by hydrogeochemical research in several deltas (e.g. Colombani et al., 2017; Fass et al., 2007; Faye et al., 2005; Manzano et al., 2001; Wang and Jiao, 2012).  The physical justification for this hypothesis, however, often still has to be tested, with a few notable exceptions (Delsman et al., 2014; Larsen et al., 2017; Van Pham et al., 2019), and can be provided by palaeohydrogeologic modelling, which has recently provided important insights for several cases. Gossel et al. (2010) created a large-scale 3D variable-density groundwater model of the Nubian Aquifer System and showed that seawater intrusion has occurred since the Pleistocene Lowstand towards the Qattara Depression (North-West Egypt). Later, Voss and Soliman (2013) showed with a

parsimonious 3D model of the same groundwater system that groundwater tables are naturally lowering during the Holocene, since they receive limited recharge and are drained into oases or sabkhas. Moreover, the authors used an inventive validation method by comparing the position of discharge areas in the model with a dataset of oases or sabkha locations. Delsman et al. (2014) conducted a detailed palaeohydrogeological reconstruction of the last 8.5 ka over a cross-section in the Netherlands to show that the system has never reached a steady state. They showed that the Holocene transgression caused substantial seawater intrusion, from which the system is still recovering. Using a combination of geophysics and 2D numerical models, Larsen et al. (2017) showed that during the Holocene transgression sea water preferentially intruded in former river branches in the Red River Delta, Vietnam. Van Pham et al. (2019) showed that most of the fresh groundwater in the Mekong Delta (Vietnam) was likely recharged during the Pleistocene and preserved by the Holocene clay cap. Despite being in a humid climate, recharge to the deeper Mekong Delta groundwater system is very limited and freshwater volumes are still declining naturally.

*Moreover, the paper as a whole, but the Discussion section in particular, is a bit of a confusing mix of a number of problems. There is (A) the scientific problem of understanding the evolution of the groundwater salinity distribution in large delta systems over long timescales. This is mixed with (B) the local management problem/question of how much freshwater there is in the Nile delta. And then there is (C) the problem that previous models have assumed steady-state conditions. For a publication in HESS, the local management problem (B) is not the most important. The scientific problem (A) is, and it should be made clear from the onset and throughout the paper that this is the main focus. The implications for the local management problem can be mentioned toward the end (it is especially interesting to note that depending on the conceptualization, the locations where freshwater occurs will differ), but should not feature prominently anywhere else.*

Following one of your suggestions in the specific comments (on P14L2), we discuss the scientific problem at the start of our discussion **[P15 L1-30]** and the local management issues to a later part of the discussion **[P16 L19-30]**. Following the suggestion in the first general comment, the scientific problem is now also discussed at the start in the introduction **[P2 L1-7]** . Furthermore, the start of the abstract is changed. This has aided a lot in changing the focus of this paper **[P1 L11-12]** .

*With regards to the assumption of the system being in steady-state (C), a considerable portion of the paper is devoted to a comparison between dynamic (i.e., models considering the palaeo-geographical evolution) and steady-state conditions (equivalent simulations for the same geology). It is concluded that the dynamic models are a better fit to the data. But given that the data set has severe limitations (there are relatively few data points, and the quality of the available data is also not assured, at least the paper does not describe the QC/QA procedures), one could wonder if that is really a such strong criterion. I think a much stronger argument could be made by looking at the time required to reach steady state. This information is not presented for all model scenarios, but in line 28 on page 11 it is mentioned that it took 60 ka for the B-model scenarios. Doesn't this automatically invalidate the steady-state assumption,*

*without having to perform anymore detailed comparisons between the dynamic models and their steady-state equivalents? I am not sure what time is required for the other simulations, but I am guessing it will be on the same order, except maybe for the most unrealistic representations of the delta's lithology. I would encourage the authors to present the timescale aspect in more detail and use it to build the argument against steady-state being a realistic assumption. Much of the detailed comparisons such as those presented in figures 9, 10 and 11 could then be omitted.*

*This would also have the benefit that the paper becomes more easy to follow, because as it stands, one quickly gets lost in the many different scenarios.*

We think the suggestion to discuss the time to the steady state is good. This, in combination with changing the order in which we discuss the results and discussion, will improve the readability of the paper. Omitting much of the detailed comparisons though, would negate one of the strong points of this study though, namely that we ran a lot of scenarios for a complex 3D model. These scenarios provide a lot of unique information that is relevant to our discussion, as it makes this study more applicable to other deltas. So here we don't fully agree. Still, we have removed what was Fig. 10 in the initial submission, as we barely discussed it in the text. Furthermore, we removed Fig. 5 and accompanying text as the data is limited.

We have added the time to reach a steady state to "3.7 model evaluation", as this section appeared to be the most fitting. We added the following lines to the methods section **[P11 L10-12]:**

To assess the validity of the steady-state assumption, we checked for all equivalent steady-state models the time they reached a steady state. This time was determined by calculating the derivative of the fresh water volume over time. If this did not change more than 1E-04% of the total volume, we considered the model to have reached a steady state.

To the results section we added **[P13 L4-8]:**

All equivalent steady-state scenarios required at least several thousands of years to reach a steady state (Table 4) from an initial Pleistocene steady state. Most notable are the B-scenarios, where the hypersaline groundwater caused the system to respond very slowly, over tens of thousands of years, thus exceeding the extent of the Holocene. The shortest scenarios were the N-T scenarios, as they did not include HGw and due to the lack of clay layers the salt water did not experience any resistance during its flow upwards from its initial Pleistocene state to the Holocene steady state.

To the discussion we added **[P15 L6-9]:**
Our equivalent steady-state scenarios required at least 5500 years to reach a steady state (Table 4), a period in which already considerable changes occurred to the boundary conditions (Fig. 3). This increased to tens of thousands of years for the more complex models. We therefore doubt that using a steady-state approach with current boundary conditions results in a reasonable estimate of the current fresh-salt groundwater distribution for such a complex, large-scale system.

To the conclusion we added **[P18 L2-L7]**:

It was found that large timescales are involved, as steady-state model scenarios required at least 5500 years to reach equilibrium. Hence, none of the evaluated paleohydrogeological scenarios reached a steady state over the last 9000 years, meaning that the transient boundary conditions definitely had an influence on current groundwater salinity. Given the large range variation in delta-architectures analyses, we can conclude that steady-state models are not likely to result in realistic FGw distributions in deltaic areas.

To the abstract **[P1 L24-L25]**::

Furthermore, the time required to reach a steady state under current boundary conditions exceeded 5500 years for all scenarios.

*The Results section*
*could start with what is now subsection 4.2, which could be expanded and/or merged*
*with subsection 4.4. This would give the reader a much better overview of the actual*
*processes before diving into the more detailed analysis of model performance and*
*freshwater volume.*

We followed this suggestion as follows: We have started the results sections with a statement that for readability we first discuss the results of a few scenarios that were selected based on a later discussed model performance assessment **[P12 L2-L5]**. Then we start with the former section 4.2 **[P12 L6]**, which describes the spatial distribution. We continue with the former section 4.4, describing the contribution of several boundary conditions through time **[P12 L14]**. After that, we discuss the model performance **[P12 L23]**.

So the outline changed to:

**4 Results**

Based on a comparison with observations, five scenarios were selected that showed the best match with the observations. These are called the "acceptable" scenarios. To keep the results comprehensible, we start with discussing the results of these five selected scenarios through space (section 4.1) and time (section 4.2), before discussing this actual selection procedure (section 4.3).

**4.1 Current spatial TDS distribution of acceptable model scenarios**

**4.2 Salt sources over time**

**4.3 Model evaluation**

**4.4 Freshwater  volume dynamics and sensitivity analyis**

5   **4.5 Fresh-salt distribution: the palaeohydrogeological reconstruction against its equivalent steady-state**

**Specific comments**
*Page 1*

10   *lines 15-16, "observed by hydrogeochemists": No need to suggest a disciplinary*
*bias*

We have removed this in **[P1 L10-L14]**.

15   *line 17, "palaeo-reconstruction": This suggests your reconstruction itself is*
*ancient. Choose better wording*

This is indeed confusing. We changed this to "palaeohydrogeological reconstruction" (just as there are for example palaeoclimate reconstructions) throughout the complete document.

*line 18: Insert "a" between "using" and "state-of-the-art model"*

We have corrected this in **[P1 L17]**

25   *line 23: You use both "sea water" and "seawater" in your manuscript. Pick*
*one, and check for consistency.*

We were indeed inconsistent and moreover grammatically incorrect in the original text. However, according to the Cambridge Dictionary "saltwater" should be used as adjective (saltwater intrusion) and "salt water" as a
30   noun.

Noun:
https://dictionary.cambridge.org/dictionary/english/salt-water
Adjective:
35   https://dictionary.cambridge.org/dictionary/english/saltwater

We assumed here that sea water and seawater should be used the same as saltwater and salt water, and hence changed "sea water intrusion" to "seawater intrusion" and use "sea water" as a noun. Thus, to stay grammatically correct, we have to stay inconsistent. The same holds for "fresh water" and "freshwater".

*line 29: add "s" to "distribution"*

We corrected this in **[P1 L29]**.

*Page 2*

*line 6: insert "that they were" between "indicated" and "pumping". More generally, the language usage needs some attention, the paper is generally well written, but every now and then it lacks some attention to detail. I will not focus on these issues from this point onward, but the authors should do a careful proofread of their revised manuscript to resolve them.*

We have corrected this specific point **[P4 L7]** and conducted extra proofreading (by a native speaker). We hope this improves our attention to detail sufficiently.

*Page 4*

*line 31: delete "hypersaline and", the fact that they are hypersaline should not be a reason to discard them. On the next page you talk about hypersaline groundwaters as well (or at least, salinities greater than seawater)*

This is indeed confusing, we have deleted these two words in **[P5 L9-10]**.

*Page 5*

*line 1: I am not sure if this reasoning holds true. A 1000y old groundwater can still move appreciable distances over a couple of decades if it is near a large well field*

Good point, our reasoning is flawed here. We have added the different measurement dates as an extra source of the spatial variation in observed salinities.

Thus changing the list of potential causes to **[P5 L11-13]**:

This variability in measured salinity can be explained with 1) the different measurement depths, 2) different dates of the measurement campaigns, 3) heterogeneity in the hydraulic conductivity of the subsoil resulting in heterogeneous salt transport, and 4) heterogeneous evapoconcentration.

*line 26: Some additional information is required here to explain the choices and rationale behind these 9 different lithological models*

We have extended this section in **[P6 L11-12].**

The height of the clayey sediments determines how disconnected the deeper groundwater system is from the sea, and thus the ability of the system to preserve denser hypersaline groundwater in its aquifers (van Engelen et al., 2018). The hydraulic conductivity of the onshore-reaching clay layers is varied to get a first-order approximation of the effect of clay layers on regional groundwater flow. We assigned a continuous hydraulic conductivity to these clay layers, based on three different lithologies (in order of decreasing hydraulic conductivity): sand, fluvial clay and marine clay (Table 3). The rationale behind this is that small clay lenses have negligible effect on regional groundwater flow, thus are assigned a hydraulic conductivity of sand. Fluvial clay layers are assigned a hydraulic conductivity of the current confining Holocene clay layer, as this was deposited under fluvial conditions (Pennington et al., 2017). Marine clay layers present continuous layers of low conductive with a big influence on the regional groundwater flow and thus have the lowest hydraulic conductivity.

*Page 6*

*line 5: Replace "its'" with "its"*

We corrected this in line **[P6 L24]**

*lines 10, 11: Not sure what you mean by "keeping memory locally".*

We meant here that each subdomain is assigned its own private memory at the computational node, instead of fitting all subdomains into one big shared memory. We changed the sentence, to clear this up. **[P6 L29-30]**

using the Message Passing Interface (The MPI Forum, 1993) to exchange data between subdomains, where each subdomain has its own private memory assigned at a computational node.

*line 15 a.f.: You need to include a map with the model area.*

We have added this map as Appendix A1. **[P40 L1-5]**

*lines 23, 24, "as the hills above this height have no important contribution to the groundwater flow": Without the aforementioned map it is hard to assess the validity of this statement. Where are these hills? How high are they? More importantly, what is the recharge and the water table elevation. The elevation of the hill itself is not so important, its hydrogeological characteristics much more so, of course...*

This was indeed an offhand remark. We do not have actual groundwater tables of this specific area, but we know that the amount of rainfall is very low and that there is no surface water located there. This is supported by Geirnaert and Laeven (1992), who found that shallow groundwater in these locations is >5000 years,

meaning very limited recent recharge. We assumed that this leads to groundwater tables well below surface level such that loss through evaporation is negligible.

We added extended this sentence to provide support **[P7 L12-14]**:

The top of the NDA was clipped off above 20 m AMSL, as the hills above this height have no important contribution to the groundwater flow (Geirnaert and Laeven, 1992), because there is very limited rainfall in the south and no surface water here.

*Page 7*

*line 3, "time slices": This sounds like what we would normally call a stress period in groundwater modelling. Why the confusing terminology? And why with a space in the title, and without in this sentence. Please pay more attention to detail.*

This is indeed what we call a stress period in groundwater modelling. Still, we decided to stick with the term "time slices", following Delsman et al. (2014). "Stress period" makes sense from a modelling perspective, but can be confusing for non-modellers (like paleogeographers, hydrogeochemists), hence why we prefer this term. We added a sentence of explanation in **[P7 L23-25]**

A time slice is also known as "stress period" by groundwater modellers (Harbaugh, Arlen, 2005)

*line 12: Replace "announced" with "at"*

We corrected this. **[P7 L32]**

Next, time slice 3 covered the marine transgression, which was a period of rapid sea-level rise at the start of the Holocene

*line 22, "100 day resistance" (better would be "a resistance of 100 days" BTW): How sensitive are the modelling results to this assumption? A single value is of course highly unrealistic, and citing studies from the Rhine-Meuse Delta does not provide any justification, because this parameter is just as uncertain and spatially (and even temporally at this timescale) variable there. But you've got to work with what you have, I understand that, but in the end, some sensitivity analysis is required to test to what extent the study outcomes might be affected by this modelling choice.*

We have conducted a local sensitivity analysis of this resistance on one scenario. We initially wanted to vary the resistance with a factor 10 or 5 but this led to numerical convergence issues for the scenarios with a decreased resistance. Therefore, we stayed at a factor 2. To not repeat ourselves in this answer, the text we added as Appendix B should explain the rest **[P18 L26]**:

To assess the effects of our assumption of the boundary resistance, we ran two alternative versions of the "H-N-T-P" scenario with a different resistance. This scenario was chosen as it presumably is the "acceptable" scenario that would be affected the most by the resistance value, as it has no horizontal clay layers that resist changes in boundary conditions and the sea boundary has the most open connection with the sea. We multiplied the resistance with a factor 0.5 and 2. Lowering the resistance more than with a factor 0.5 lead to numerical convergence issues. Fig. B1 shows that throughout the Pleistocene the resistance influences the groundwater types, as the lower resistance allows more river water to be replaced with hypersaline groundwater. The groundwater types of the different models quickly converge, however, through the Holocene. We therefore think that the choice of boundary resistance has limited effect on our results and conclusions, despite that we only varied the resistance to a limited extent in this sensitivity analysis.

*line 31: replace "acquiring" with "achieving"*

Corrected in **[P8 L20]**.

*Page 8*

*lines 3-6: Did you do this via SEAWAT's density options in the CHD package?*
*A range of 2-18 g TDS/L gives quite a range in density. What value did you adopt? And*
*again, how sensitive is it?*

We used the RIV package for the lagoons. We estimated the mean salinities from the Flaux et al. (2013) *A 7500-year strontium isotope record from the northwestern Nile delta (Maryut lagoon, Egypt)* for each time slice. Of the total of seven time slices, the last four have lagoons, since lagoons started to form from 7.5 ka. To be specific, the salinities assigned to time slices 3 to 7 are respectively: 18, 9, 4.5, 2.5 g TDS/l. Despite these decreasing salinities, we can still observe quite high salinities at the (former) locations in lagoons that fall in between 5-15 g TDS/l (see Figure 4), so past lagoonal salinities are still locally present in our model.
We do not, however, think that different salinities will have a large effect on our conclusions. Since the mixing zones are relatively small, changing the lagoonal salinity will only have a small effect on the fresh groundwater volumes, as long as this salinity does not get lower than 1 g TDS/l. This is presumably more controlled by the location of the lagoons. Which model scenarios were deemed "acceptable" was in the end mainly assessed based on the location of hypersaline groundwater, so this will also not change strongly, since the density of HGw is much higher (up to 120 g/l) than that of the lagoons (up to 18 g/l). This brackish groundwater will exert limited force on the HGw.

We added the following text that is more specific here **[P8 L23-27]**:

Of the total of seven time slices, the last four have lagoons, since lagoons started to form from 7.5 ka. The lagoon stage was set such that it was in hydrostatic equilibrium with the sea, so that its pressure is corrected for salinity (Post et al., 2007). Lagoonal palaeo-salinities were estimated from the published strontium isotope ratios from the Maryut lagoon (Flaux et al., 2013) for each time slice. To be specific, the salinities assigned to

time slices 3 to 7 are respectively: 18, 9, 4.5, 2.5 g TDS/l. These salinities show strong variation through time, because the inflow of the Nile varied through time.

*Page 10*

*line 7: You could cite the following article here: Sanford, W.E. & Pope, J.P. Hydrogeol J (2010) 18: 73. https://doi.org/10.1007/s10040-009-0513-4*

This is a very fitting suggestion, thanks. Also relevant for the discussion.

We added the reference to method section 3.7 **[P10 L29]** and added an extra sentence to the discussion **[P16 L1-2]**:

Similar conclusions were drawn by Sanford and Pope (2010) for the Eastern Shore of Virginia (USA), an area with a similar observation density.

*line 10 a.f.:*
*This is somewhat hard to follow and it might be worth adding a sketch that illustrates the principle (could be in a supplementary document). Other authors may choose to adopt this methodology, hence it could be worthwhile doing this.*

We have added a sketch as Fig A2 **[P41 L1-5]**.

*Page 11*

*line 19: what do you mean with "behavioural"? And what is the justification for using 0.07?*

With the term "behavioural" we mean the ability of models to reproduce certain patterns observed, following Beven and Binley (1992) *The future of distributed models: Model calibration and uncertainty prediction.* Whatever these patterns specifically are, is up to the researcher to decide. So, on second thought, perhaps the word "acceptable" captures the inherent arbitrariness of this decision better. The value of 0.07 was based on a visual inspection of the figure, as around that value there is a separation visible. The scenarios with $Md|\Lambda| < 0.07$, predict the location of the HGw with similar skill as to with which they predict the location of saline groundwater. These scenarios we call "acceptable" .

In the text **[P12 L29 – P13 L3]**:

More striking differences are observed in the hypersaline zone, where we observe a division around $\Lambda = 0.07$ into two groups. There are the scenarios with $Md|\Lambda| < 0.07$, that predict the location of the HGw with similar skill as to with which they predict the location of saline groundwater. We call these scenarios "acceptable". Specifically, these are the following five model scenarios: C-M-B-P, C-N-T-P, H-M-T-P, H-F-T-P, H-N-T-P. The other scenarios perform considerably worse in predicting the location of the HGw.

*lines 19-21: Sentence does not seem to flow well due to a grammar error,*
*not sure what you are trying to say here.*

This should be clear now **[P12 L29 – P13 L3]**.

*Page 12*

*line 3: Not clear what you mean here with "behavioural"*

Hopefully cleared up now with the changes made to the description of the model evaluation **[P12 L29 – P13 L3]**. See response comments of P11L19.

*line 4: Up until*
*this point the difference between fluvial and marine clay layers has not been explicitly*
*discussed.*

Now expanded on in section 3.1 **[P6 L11-12]**. So see answer to comment for P5L26.

*lines 7, 8: replace nondescript terms like "more 3D patterns are visible in*
*the salinities" and "partly has a conical" with more accurate descriptions*

We are more specific here now **[P12 L10-13]**

Regardless of the differences between scenarios, in all realizations the fresh-salt interface roughly follows the coastline, except in the west where there is far extending seawater intrusion visible towards Wadi El Natrun (Fig. 1). Next to this depression, (former) lagoons are visible as shallow brackish zones and (former) dune areas are visible as freshwater lenses.

*lines 24 a.f., "This table also shows...": I could not follow what you are trying to say here*

This sentence could be more to the point. Changed the sentence to **[P13 L22-L24]**:

This table also shows that these parts of the model are also the most uncertain, since disregarding potential deep and offshore fresh groundwater volumes decreases the spread from 74% to 32%.

*Page 14*

*line 2 a.f.: I think the point you want to make here is that you can come up with*
*multiple models that fit the observations equally well and yet, the volume of freshwater*
*varies a lot. What do you mean with "The variance in the results should also affect*
*management decisions."? I think the management decisions will not be based on the*
*total volume of freshwater, but on the possibility to be able to extract groundwater in*
*a particular region. See also general remark made before about the relevance of the*

*freshwater volume issue for this paper. I would not start the Discussion section with*
*this paragraph (see next point)*

We have removed these sentences now, since it shifts the focus too much to the regional problem. We tried to say that policy makers and managers should take into mind the large uncertainty of groundwater models in these complex aquifers, as shown in this research, but since we do not provide any suggestion to do so this is a bit of a weak statement.

However, even though it has no effect on the changes made to this paper, we would like to express our different view on how management decisions are made in the Nile Delta though. Currently, there is an ongoing political discussion between Egypt and the upstream countries on the effects of the large dams that are being built. Discharge of the Nile will decrease and thus the volume of fresh groundwater that can serve as a "strategic stock" during periods of low-flow is becoming of increasing interest, from the perspective of delta-scale management.

*Page 15*

*line 6: Start the Discussion section with this paragraph, it is much more relevant*
*to a broad readership than local aspects such as the discussion of the total*
*freshwater volume (also see general comment) and flow to Wadi El Natrun.*

Good suggestion, we have brought this paragraph to the start of the Discussion section. **[P15 L2-30]**

*line 32: Also include a discussion of the representation of free convection phenomena*
*in your model. The large grid size is prohibitive for an accurate process representation.*
*How confident are you that this does not harm the general conclusions drawn from the*
*model outcomes?*

We are confident that the general conclusions are not harmed, since we have investigated the errors made in modelling with a crude resolution for the Nile Delta in a previous paper. This was published as an appendix, so easily overlooked. We have added a deliberation on this in the discussion of this paper in lines **[P17 L10-18]**. For your convenience we have added this appendix of our previous paper as supplementary information to this response to authors.

To the discussion we added:
In addition, for a proper physical representation of free convection, a finer grid is required. A coarse horizontal cell size results in a delay in the onset of free convection, while a coarse vertical cell size results in an onset of free convection even for situations that are expected to be stable (Kooi et al., 2000). van Engelen et al. (2018, Appendix D) investigated the errors caused by coarse model cells for the Nile Delta and found that especially the crude horizontal grid size had an influence. They found that this resulted in similar downward fluxes, but a delay in the onset of free convection. This effect, however, was negligible after ~50 years and thus dwarfed by the timescale of our time slices. The coarse vertical grid size was not an issue, since the marine transgression occurred over sand with a high hydraulic conductivity, meaning there is a very instable situation and free

convection has to occur. We thus think that the errors made in modelling free convection do not influence our conclusions.

*Figure 1: It could just be because of the pdf, but the resolution is very poor. Figure needs an inset showing the location of the area within Egypt/the Mediterranean region. Add north arrow and scale bar! In this figure all areas outside the delta are white, better to make the Mediterranean blue and the desert yellow(-ish).*

Our maps indeed lacked these crucial map features, so therefore we added them to this figure **[P30 L1-2]**. The resolution presumably deteriorated in converting the vector file first to a .png to add in MS Word and consequently to a pdf. We plan on submitting the original vector files as pdfs for the final version, which should solve this.

*Figure 2: Somewhere we will need a map with the model boundary. On this map you will need to indicate which part of the model is shown here.*

We added a dedicated figure to this (Fig. A1), since adding these features to Fig. 1 resulted in a too complicated, cluttered map.

*Figures 4, 5: Take the reader by the hand here as the figures are complex. Explicitly mention in the caption that a, b, c and d reflect the salinity classes, and explain the n value.*

We changed the caption to **[P35 L1-8]**:

Figure 6: Goodness of fit boxplots for all palaeohydrogeological reconstructions, binned into four salinity classes. The higher the value of $\Lambda$, the worse the fit. Codes indicate model scenarios (Table 1). TDS values are binned in the classes [0, 1], [1, 5], [5, 35], [35, 100] g TDS/L for respectively "fresh", "brackish", "saline", "hypersaline", these are respectively plotted in panels A, B, C, D. "n" indicates the amount of observations available in the TDS bin to evaluate model results with. Diamonds indicate outliers, defined as values separated from the first or third quartile at 1.5 times the interquartile range. When no box is plotted for a scenario, 75% of the measurements equal zero, which consequently causes the interquartile range to be zero, rendering every non-zero value an outlier.

**Supplementary material Answer to Reviewer 1**

**Effects grid size on free convection**

**Context:**

In this Appendix we conducted a grid convergence test for a simple sandbox model. "NDA-f model" means "Nile Delta Aquifer free convection model". There was also a model without free convection, and for this model we conducted a grid convergence test on the original model. Therefore, the text below starts with the words "different approach". In addition, we removed all less relevant parameters to this research from Table 1.

The text below is taken from Appendix D of :
**van Engelen et al. (2018)** *On the origins of hypersaline groundwater in the Nile Delta aquifer*

For the NDA-f model, a different approach was taken, to assess the effect of a wider range of $\Delta x$, since this dimension had the coarsest resolution (1 km). We modeled a simple 800 m thick sandbox with a hypersaline lake on top. This allowed us to stretch the domain as much as necessary in the horizontal direction. The chosen parameters for this model are the same as in Table 1. The amount of cells was kept at 100 in the horizontal direction. $\Xi$ is the normalized fluid mass increase, which is calculated by dividing the increase in fluid mass across the top boundary by the maximum mass storage increase:

$$
\text{(D.1)} \qquad \Xi = \frac{\iint \frac{Q}{\Delta x} \rho \, \mathrm{d}x \mathrm{d}t}{V_{tot}\left( (\rho_{max} - \rho_0)n + \frac{1}{2} S_f \frac{\rho_{max} - \rho_0}{\rho_0} z_{max} \rho_{max} \right)}
$$

where $Q$ is the volumetric flux through a boundary cell (m²/d), $V_{tot}$ is the total volume of the model domain (m²), $\rho_{max}$ is the concentration at the top boundary which was set at 1078 kg/m³, $\rho_0$ was the initial density of the groundwater in the domain (1000 kg/m³), $S_f$ is the specific storage in terms of fresh water head (d⁻¹), $z_{max}$ the depth of the domain (m), and $n$ the porosity (-). For the test with $\Delta x$, $\Delta z$ was kept at 10 m; in testing $\Delta z$, $\Delta x$ was kept at 10 m. As the density at the top boundary of the model was perturbed randomly, 10 simulations were started for each discretization, of which the minimum, maximum and mean of $\Xi$ are plotted. It can be seen in figure D.2 that there is some spread in $\Xi$, which is caused both by integration errors and differences between realizations. Furthermore, a larger $\Delta x$ causes a delay in the onset of free convection and causes a slower downward movement of fingers. However, all errors are unimportant on timescales larger than ~100 years, and thus also on our timescales of interest, which is over 1000 years.

*Table 1. Parameters for the NDA-f model. We are purposely inconsistent with units, as it allows for easier comparison with values found in the literature.*

| Parameter | Description | Value | Unit |
|---|---|---|---|
| $K_{h,\,sand}$ | Horizontal hydraulic conductivity sand. Compaction case and free convection case | 75 | m/d |
| $\dfrac{K_h}{K_v}$ | Anisotropy | 10 | - |
| $\alpha_l,\ \alpha_t,\ \alpha_v$ | Longitudinal, transversal and vertical dispersion length | 10, 1, 0.1 | m |
| $n_e$ | Effective porosity | 0.10 | - |
| $\dfrac{\Delta h_{riv}}{\Delta x}$ | River gradient | 9.375e-2 | m/km |
| $S_f$ | Specific storage in terms of fresh water head | 1e-5 | 1/m |
| $\dfrac{\partial \rho}{\partial C}$ | Slope linear equation of state | 0.71 | (kg/m$^3$)/(g/l) |

[Figure]

*Figure D.2: Results of the test of the errors that are introduced in the free convection model by the large grid size: a. shows the influence of Δx and b. shows the influence of Δz.*

**Answer to Reviewer 2**

We thank reviewer 2 for his/her thoughtful review that greatly helped to improve our paper. In this document, we have entered your comments in italics, added our response as regular text, and subsequently added suggested changes to the paper in red. After our responses you can find the suggested changes made to the text in red. Line numbers are written as "**[P…L…]**", indicating page number, line number in the no markup document.

**General comments**

*General comments: This paper describes numerical modelling used to reconstruct the salinity distribution in the Nile Delta aquifer over the past 32 ka. The model simulations include the Holocene marine transgression and various scenarios relating to possible aquifer lithology, and sources of hypersaline seawater. Through consideration of palaeo-environments, the authors aim to better represent the Nile Delta aquifer and an observed freshening. The authors state that simulation of the freshening of the aquifer has not been possible previously using realistic model parameters and steady-state modelling. The objectives of this modelling study are stated as being (p4, L8) to: 1) investigate the physical plausibility of the Holocene transgression hypothesis for the Nile Delta; 2) investigate the influence of the uncertain geology; 3) provide estimates of the present-day fresh groundwater [FGW] volumes; 4) assess the importance of using palaeo-reconstructions compared to less expensive steady-state modelling. These objectives have been addressed within the study and the description of how this was achieved is generally clearly detailed. There are not very many palaeo-reconstruction type studies looking at saltwater-freshwater dynamics in coastal aquifers and therefore there is some novelty in this contribution. However the significance is not clear. The study relates specifically to the Nile Delta aquifer and the more generalised extension of the science is not apparent. As such the study is primarily a case study, albeit with some interesting insights for other similar deltaic systems. For this manuscript to be suitable for publication in HESS the authors need to indicate how the study provides a substantial contribution, and is not just a case study.*

We think with the help of also the general comments of Reviewer 1, we managed to change the focus of this paper from the regional problem to the scientific problem. We start the introduction **[P2 L1-7]** and abstract now with the general scientific problem. Furthermore, we discuss the scientific problem now at the start of our discussion **[P15 L1-30]** and the local management issues are kept as a later part of the discussion **[P16 L19-30]**. Furthermore, the start of the abstract is changed **[P1 L11-12]**. This adaptation should more clearly aid in stressing the scientific significance of this research.

Furthermore, given the wide range of lithological scenarios, several of the findings are also applicable to other deltas: The severe decrease in FGw volume after the marine transgression (at the start of the Holocene), the large variation between the different palaeohydrogeological reconstructions compared to steady-state scenarios, and the lithologies that favor offshore fresh groundwater volumes in deltas.

To show this stress this more, we added this to the conclusion **[P18 L6-L9]**:

Given the large range variation in delta-architectures analyses, we can conclude that steady-state models are not likely to result in realistic FGw distributions in deltaic areas. Our results also show that the occurrence of past marine transgressions constitute a valid hypothesis explaining the occurrence of the extensive saline zone land inward.

**Specific comments**

*Specific comments: The use of iMOD-SEAWAT has presumably assisted in reducing*
*run times and making modelling of the various scenarios used in the study tractable.*
*The paper would benefit from more information describing the run times and benefits of*
*using iMOD-SEAWAT. I expect that other researchers will find this interesting for similar*
*palaeo-reconstruction type work.*

The added benefits of this code were already described (namely faster runtimes on computers with multiple CPUs), we however could add some information on run times **[P6 L32 – P7 L1]**.

Simulations were conducted on the Dutch national computational cluster Cartesius (Surfsara, 2014), using Intel Xeon E5-2690 v3 processors. With this new code the model scenarios had a wall clock time ranging of 44 hours to 108 hours on 48 cores, depending on model complexity.

*The study involved numerous scenarios and I found many of these confusing as I*
*read through the paper. The section describing the coding of the scenarios needs*
*improvement. For example, I was not able to discern the meaning of P in the scenario*
*H-F-B-P (p10, L1), despite that this was used as part of the explanation of how the*
*coding system works.*

We can understand that this might be confusing, as it was very difficult to come up with an appropriate coding system. To reduce the confusion we expanded the example and added each feature-member combination behind each piece of the sentence where it was specified **[P10 L19-23]**:

For each feature, the corresponding letters in table 1 are converted to a code as follows: *{sea}-{clayer}-{prov}-{temp}*. For example, the palaeohydrogeological reconstruction (*temp*: P) of an aquifer with a half-open sea connection (*sea*: H), fluvial horizontal clay layers (*clayer*: F), and HGw seeping in from the bottom (*prov*: B) gets the following code: H-F-B-P. Its equivalent steady-state model is attributed the code H-F-B-S.

*Additionally, the term 'behavioural' model scenarios (p. 12, L1)*
*seems odd and I suggest changing.*

Copying from the response to comment to P11L19 by Reviewer 1:

With the term "behavioural" we mean the ability of models to reproduce certain patterns observed, following Beven and Binley (1992) *The future of distributed models: Model calibration and uncertainty prediction.* Whatever these patterns specifically are, is up to the researcher to decide. So, on second thought, perhaps the word "acceptable" captures the inherent arbitrariness of this decision better. Hence, we have changed the word "behavioural" to "acceptable" in the paper.

In the text **[P12 L29 – P13 L3]**:

[revised manuscript text omitted]

8 ka

Salinity

**TDS (g/L)**
- ▲ 0 - 1
- ▲ 1 - 5
- ▲ 5 - 15
- ▲ 15 - 25
- ▲ 25 - 35
- ▲ 35 - 55
- ▲ 55 - 75
- ▲ 75 - 100

**Depth (m)**
- ▲ 50
- ▲ 100
- ▲ 150
- ▲ 200
- ▲ 250

**Data Source**
- ● 80's campaign in Laeven (1991)
- ■ 90's campaign in Laeven (1991)
- ▲ RIGW (2013)
- ⬠ Geriesh (2015)
- ◣ Salem (2016)

7.5 - 6 ka

Maryut lagoon

Paleogeography
- ▢ Lagoons
- ▢ Coastal Marshes
- ▢ Holocene Clay (Bilqas 2)
- ▢ Coastal barrier-breaches/Strand plains
- ▢ Pleistocene Sands (Mit Ghamr & Gezira cover)
- ▢ Sabhkas & Carbonate ridges
- ▢ Wadi El-Natrun
- ⋯ Current Coastline

[revised manuscript text omitted]

---

## Referee Report (RR1)

[referee-annotated manuscript omitted]

---

## Referee Report (RR2)

[referee-annotated manuscript omitted]

---

## Author Response (AR2)

**General response authors**

We thank Reviewer 1 for his thorough and constructive review. They definitely improved the readability of this manuscript. We agree with most remarks. Below we only listed the responses to comments (highlighted in yellow in the pdf), as we have accepted all the suggested small textual changes (That were crossed out in blue or red in the pdf).

Replies to comments are structured as follows: First, we list the sentence to which the remark refers in quotes, preceded with **P…L…** Second comes the reviewers remark to this sentence in italics. Third comes our reply to these remarks. Finally, if relevant, how we adapted the text in red.

In our responses line numbers are written as **"[P… L…]"**, indicating page number, line number in the no markup document.

Two changes affect multiple locations in the manuscript: 1) Our units now conform to the HESS standards and 2) the word "time slice" is changed to "stress period".

Furthermore, we included the following sections that were missing in the previous version of the manuscript: "Code and data availability section", "Author contribution", "Competing interests"

**Specific comments**

**P2L2**: "since groundwater below 250 m depth is dominated by groundwater with an age of over 12 ka (Jasechko et al., 2017). "

*There are much better studies you can cite here. The 250 m immediately introduces some arbitrary criterion and large-scale is not the same as deeper than x m. Better refer to the Palaeaux monograph by Edmunds et al or something.*

We agree that the 250 m creates arbitrariness and that large-scale groundwater system is not the same as a system with a large depth. However, there is a strong correlation between the depth of a groundwater system and its' scale, therefore we think the reference to Jasechko et al. is still fitting as he shows old groundwater globally, not just in Europe or the US.
We therefore deleted the subclause that introduced arbitrariness and we also included the suggested reference to Edmunds et al. as the PALEAUX project goes into more detail on the palaeohydrogeology for several large aquifers in Europe **[P2L2]**.

Palaeohydrogeological conditions have influenced groundwater quality in the majority of large-scale groundwater systems (Edmunds, 2001; Jasechko et al., 2017).

**P2L33**: "despite that the aquifer was gaining fresh water at that time."

*Not sure what you are trying to say here. It seems that you want to make the point that Kashef's hypothesis runs contradictory to hydrochemical data, but this is not clear*

We tried to state here that these two points are contradictory in the following sense: the aquifer gaining fresh water is an argument to allow more groundwater pumping, whereas the seawater wedge reaching far inland is not. We extended the subclause **[P3L1-P3L3]**:

As an example of a study based on groundwater mechanics, Kashef (1983) used the Ghyben-Herzberg relationship to show that the seawater wedge reached far inland, thereby showing that the NDA should be carefully managed despite that the aquifer was gaining fresh water at that time.

**P3L7**: "overlooked"

*Or ignored?*

Given that the data for these depths is hard to find, we do not think that the previous studies intentionally ignored this.

**P3L11**: "groundwater mechanical"

*mechanical does not sound good here. It sounds too much like geomechanical, geotechnical, compaction, land subsidence, that kind of thing. The Domenico and Schwartz book uses phyiscal to contract chemical, perhaps that is better. Or use "flow"*

For our initial submission we had considered several options after which we ended up with "groundwater mechanical", which is a term used by O.D.L Strack for his book "Groundwater Mechanics". This book is well known, though not as much as Domenico & Schwartz' book (e.g. it has been currently cited nearly 1000 times compared to nearly 5000 times for Domenico & Schwartz according to Google Scholar.)
"Flow" is not a fitting term here as it does not cover the Ghyben-Herzberg approximation, which is purely hydrostatic. "Physical" is a broader term than "mechanical", so potentially even more confusing. For example, "Physical groundwater studies" can also imply the opposite to "Virtual groundwater studies", e.g. https://doi.org/10.1073/pnas.1500457112. Therefore, we decide to stick with "Groundwater mechanics".

Nevertheless, to prevent confusion, we added the following sentence to the first mentioning of "groundwater mechanics" **[P2L28]**:

The latter subdivision covers all studies focused on groundwater flow and statics, following Strack (1989).

**P3L12**: "Many"

*Be more specific. Which ones did, which ones did not and what did they do exactly if they did not completely ignore the palaeohydrological conditions.*

We are more specific now **[P3L14-P3L16]**:

All the previously discussed groundwater-mechanical studies except Van Engelen et al. (2018) neglected the influences of palaeohydrogeological conditions that were inferred by hydrogeochemists.

**P3L15**: "large influence of hydrodynamic dispersion"

*You are somewhat repeating the reasoning that you used in the previous paragraph. It seems that both paragraphs need some restructuring, merging and fine-tuning.*

We have edited this sentence as there was a minor repetition here, but we disagree that we are repeating the same reasoning in this paragraph as in the previous one. The previous paragraph only explained what previous studies found, even avoiding mentioning of the Holocene transgression. In this paragraph we compare the two study fields and identify a discrepancy between them. We do not really see how we are repeating ourselves here, apart from this specific point on dispersion. It could be that it feels like we are repeating ourselves because the abstract is quite close to this paragraph, because here we make this argument on the large dispersivities. We have decided to only make minor changes, but not to restructure and merge these two paragraphs as it would not contribute to the clarity of the text **[P3L16-P3L17]**:

Instead, they used longitudinal dispersivities exceeding 70 m (Sefelnasr and Sherif, 2014; Sherif et al., 1988) to simulate the large brackish zone in the groundwater system (Fig. 1)

**P3L17**: "and the extent of this brackish zone coincides with palaeo-shorelines (Stanley and Warne, 1993) (Fig. 1). "

*Why don't you also cite https://www.cambridge.org/core/journals/netherlands-journal-of-geosciences/article/geological-processes-and-the-management-of-groundwater-resources-in-coastal-areas/A62ABC242EA271A6CC2C415E938CCC11/share/999729834d9ef57f3a76c3624b5d1f7c242a847c*

We have added this reference **[P3L19-P3L20]**:

and the extent of this brackish zone coincides with palaeo-shorelines (Kooi and Groen, 2003; Stanley and Warne, 1993) (Fig. 1)

**P3L31**: "2) investigate the influence of the uncertain geology;"

*See (and cite) also: https://www.sciencedirect.com/science/article/pii/S0022169418309387*

Thank you for this suggestion, this review indeed covers what we did: testing the Conceptual Physical Structure. We have added the reference **[P3L34]**:

2) investigate the influence of the uncertain geology (Enemark et al., 2019)

**P4L29**: "Hydrogeological research dealing with this area generally assumed"

*References are required to back up this statement*

This was indeed an offhand remark, we have added references **[P4L31-P5L2]:**

Hydrogeological research dealing with this area generally assumed that the connection between aquifer and sea is completely open (e.g. Kashef, 1983; Mabrouk et al., 2018; Sefelnasr and Sherif, 2014; Sherif et al., 1988)

**P5L1**: "are more commonly"

*more common than what, or where...?*

Changed to **[P5L4-P5L5]**:

These observations can be explained by the fact that the fraction of marine clays generally increases seawards (Nichols, 2009).

**P5L11**: "This variability in"

*Could it also be poor data quality? If so it is fine to acknowledge this.*

This could indeed be true, since the data quality assurance was not always clear in the reports. We have therefore changed point 2 of the following list into **[P5L16]**:

2) different data sources, with different data quality and dates of their measurement campaigns

**P5L20**: "becomes apparent"

*hard to know for the reader what you exactly mean here...*

We are more specific now **[P5L24-P5L26]**:

A comparison of our dataset with soil salinity maps (Kubota et al., 2017) shows that in the coastal areas soil salinities are high, which are the areas where we have the least measurements. This bias is particularly evident from the data gathered by Salem et al. (2016b).

**P6L7**: "is covered completely with low-permeable clayey sediments or not"

*you present a choice here between completely covered or not (at all). Why can't it be partially covered? Nature is not often binary...*

You are right, sentence is changed to **[P6L6-P6L7]**:

to what extent the continental slope is covered with low-permeable clayey sediments.

**P6L19**: "Marine clay layers present continuous layers of low conductivity with a big influence on the regional groundwater flow and thus have the lowest hydraulic conductivity."

*sentence makes little sense as you are saying that layers of low conductivity have the lowest conductivity*

You are right, sentence is changed to **[P6L18-P6L19]**:

Marine clay layers present continuous layers of low conductivity with a big influence on the regional groundwater flow

**P7L23**: "A time slice is also known as "stress period" by groundwater modellers (Harbaugh, Arlen, 2005)."

*I really don't see the need to introduce a new term from something that is well understood within the scientific community. It only creates confusion in our terminology.*

OK, we have changed this throughout the paper and updated figures 5, 7, and B1.

**P7L24**: "consisted"

*Not the right word.*

Changed to **[P7L23]**:

Each stress period was assigned a palaeogeographic map

**P8L27**: "These salinities show strong variation through time, because the inflow of the Nile varied through time. "

*Do you mean that the decrease in time you describe in the previous sentence was caused by increasing Nile water inflow?*

The explanation is indeed not clear. We have improved the level of specificness in this paragraph **[P8L26-P8L29]**. To alleviate doubt this might create we quote our previous author comment: "We do not, however, think that different salinities will have a large effect on our conclusions. Since the brackish zones are relatively small, changing the lagoonal salinity will only have a small effect on the fresh groundwater volumes, as long as this salinity does not get lower than 1 g TDS L$^{-1}$. This is presumably more controlled by the location of the lagoons. Which model scenarios were deemed "acceptable" was in the end mainly assessed based on the location of hypersaline groundwater, so this will also not change strongly, since the density of HGw is much higher (up to 120 g L$^{-1}$) than that of the lagoons (up to 18 g L$^{-1}$). This brackish groundwater will exert limited force on the HGw. "

The decreasing trend in salinities is partly the result of a progressively humid climate and increased human influence (Flaux et al., 2013) and partly the result of averaging over our stress periods. From 4 ka to 3 ka there was an arid period with high salinities (~18 g TDS L$^{-1}$), but since we based our stress periods on the available palaeogeographic maps, this spike is dampened by the preceding 2 ka period of brackish conditions (~5 g TDS L$^{-1}$).

**P9L14**: "including extraction wells (locations and rates)"

*Provide some key figures here, for example the total annual abstraction and how it has increased over time*

We have added a figure to the appendix (Fig. A2), since the focus of this paper is on the palaeohydrogeology and not as much on the human influence.

**P9L24**: "d$^{-1}$"

*Flux should be length per unit of time so presumably m/d? Also note that you are mixing two different notation styles for units here. Choose one and be consistent (this comment was also made on the original version of the manuscript and still stands).*

Yes, we made this mistake because the source term in the governing equation in the Modflow manual is in d⁻¹, but as we discuss a flux here it should be in m d⁻¹. The value has changed as we multiplied the value in d⁻¹ with the thickness of model cells at the bottom (50 m).
We have furthermore conducted a thorough check on units to ensure they conform to the HESS standards, e.g. m2/d is changed to $m^2 d^{-1}$.

**P9L30**: "completely fresh"

*Comment on the assumption that all salt from the previous transgression being flushed out is reasonable.*

The scenario where we would expect the most salt-water to be preserved from previous transgressions are the "C-M" scenarios (Closed off sea, very low permeable horizontal clay layers). In the C-M-B-P scenario, we started with an initially salt aquifer (result from van Engelen, 2018). In this scenario we observed that during the Pleistocene the top half was completely flushed to fresh in 80 ka. The bottom half stays hypersaline because it receives hypersaline groundwater from the bottom (compaction-induced flow). Given that the top half of our models is where most variability occurs in salinities, we are therefore convinced that due to the high hydraulic conductivity of the sand in the NDA, the choice of initial conditions has limited effect on our results and no effect on our conclusions.

**P10L8**: "the bottom of the aquifer should be closed off."

*unclear, please explain*

We have removed this sentence and added more clarification two sentences further **[P10L14-P10L17]** (see next comment).

**P10L11**: "could be"

*Do you believe that these are physically unrealistic, or was it purely a pragmatical choice because of convergence issues? Or both?*

These three scenarios were physically unrealistic. It was a pragmatical choice in the sense that less numerical simulations had to be run, but we think they would converge as no isolated bubbles of fresh water would form in between clay layers when there is an open sea boundary. We agree that this sentence was confusing, so we are more specific now **[P10L14-P10L17]**:

Based on previous findings in Van Engelen (2018), three combinations could be excluded, as these would not produce the observed salinity distributions. These were the combinations with an upward HGw flux originating from the bottom and with a completely open sea boundary, which lead to all HGw to be immediately drained into the sea resulting in only a very small volume of HGw in the NDA.

**P10L11**: "These were the combinations where HGw was coming from the bottom and the sea boundary was completely open."

*This reads like a repetition of the sentence in lines 9 and 10.*

This indeed got repetitive, we have removed a repetitive sentence that was in lines 8 and 9.

**P10L14**: "Pressures in these zones increased fast"

*During sea level rise?*

Correct, we have added this clarification **[P10L19-P10L20]**.

Pressures in these zones increased fast during sea-level rise

**P12L2**: "acceptable"

*What about plausible?*

We have considered this word, but we preferred "acceptable" as we think it covers better the inherent arbitrariness of the choice of models with a HGw zone that satisfy $\mathrm{Md}|\Lambda| < 0.07$.

**P12L8**: "The model scenarios without clay layers (N) or fluvial clay layers (F) have freshwater distributions very similar to the O-N-T-P model scenario."

*N = no clay, so this sentence sounds weird as you are saying that "... model scenarios without clay layers .... have freshwater distribution very similar to the " O-(no clay)-T-P scenario.*

You are right, we are more specific now **[P13L7-P13L8]**:

The model scenarios H-F-T-P, H-N-T-P, and C-N-T-P, which have no or fluvial clay layers, have freshwater distributions very similar to the O-N-T-P model scenario.

*I wonder why H-F-T-P and H-N-T-P have such similar salinity distributions and freshwater volumes, even though the fluvial clays have a much lower conductivity than sand.*

The differences are indeed not very noticeable, the H-F-T-P model does respond a bit slower which can be seen in the first stress period in the fresh water volumes (Fig 7a) and furthermore its salinity distributions are more chaotic (Fig 4, bottom). We attribute the only slim differences in the Holocene to the fact that the timescale of the model is large enough to cancel out the slower response caused by the fluvial clays. If the boundary conditions in our model changed more frequently, we think the differences would be more pronounced than they are now.

**P15L1**: "Discussion"

*Also discuss your findings in relation to the findings by Michael et al (2016): https://agupubs.onlinelibrary.wiley.com/doi/full/10.1002/2016GL070863*

We have added the following sentences to **[P16L12-P16L15]**:

Michael et al. (2016) showed with steady-state models that strong, well connected heterogeneities can lead to chaotic salinity distributions, showing a high spatial variability that we also observe in our most heterogenous cases (Fig. 4). This variability only increases when palaeohydrogeology is accounted for (Fig 7b), because there is less time for mixing to smoothen strong concentration gradients.

**P15L14**: "or as a steeper fresh-salt interface than the steady-state interface, which to the authors' knowledge has never been shown before in this detail."

*This may be apparent in your models, but how are you ever going to demonstrate this with field data? After all, you can never measure the slope of the steady-state interface as it does not exist.*

True, we have modified this sentence **[P16L15-P16L17]**:

Furthermore, we have shown that it is physically possible that the influences of the Holocene marine transgression can be still observed in the delta; either as freshwater volumes in between clay layers in the northern part of the NDA, concurrent with earlier research (Kooi et al., 2000), or as a steeper fresh-salt interface than would be expected a priori from exploratory steady-state models.

**P15L17**: "it"

*Not clear what "it" is referring to here.*

"it" refers to the non-steady interface. We have clarified this now **[P16L21-P16L23]**:

Likewise, this non-steady, steep interface could explain the observed freshening of the shallow NDA as observed in hydrogeochemical studies (Barrocu and Dahab, 2010; Geriesh et al., 2015), since only the zone above 250 m depth has been sampled.

**P15L19**: "since only the zone above 250 m depth has been sampled"

*It is hard to follow your argumentation here. It sounds as if the freshening has something to do with the steady-state interface.*

We have added a sentence with clarification here **[P16L23-P16L24]**:

… since only the zone above 250 m depth has been sampled. This is the zone that we expect to slowly freshen from our comparison between palaeohydrogeological and steady-state models (Fig. 9).

**P15L20**: "with model results "

*But from the model you can infer what process causes freshening. This is now not discussed.*

This is a good suggestion that increases the relevance of the paper. We have added a sentence on this and also replaced a vague "this" into "the observed freshening" **[P16L24-P16L27]**:

It was however difficult to compare the observed freshening with model results directly, since the time scale over which this observed freshening has developed is often unknown (Stuyfzand, 2008). We can

affirm though that it is physically possible that the NDA was predominantly freshened with surface water (Fig. 5), as hypothesized earlier by Geirnaert and Laeven (1992).

**P15L32**: "This variation is mainly attributed to"

*This is not true. The variation is due to the different model setups, it has nothing to do with the observations.*

This is indeed written in a confusing way. We have changed the previous sentence for clarification **[P17L3-P17L4]**:

A wide range of model scenarios resulted in acceptable results. This is mainly attributed to the limited availability of salinity observations in the saline zone, which is the area where the results of the acceptable scenarios differed the most.

**P16L3**: "Still, the FGw volume estimates are on the large side."

*Do you mean your model-based esitmates? Don't start with "Still", it is confusing.*

OK, we have changed the sentence to **[P17L8-P17L9]**:

Our model-based FGw volume estimates are on the large side.

**P16L11**: "as the current study also incorporated a steep glacial hydraulic gradient, whereas van Engelen et al (2018) kept a constant low (interglacial) hydraulic gradient."

*The meaning of this sentence remains unclear until one reads the entire paragraph. I would therefore reverse the order, first describe what you observe in the models, and then state that Van Engelen et al (2018) could not have come to the same conclusion because they did not consider the steep gradient scenario*

We have followed your suggestions, as it improves readability **[P17L15-P17L23]**:

Apparent from the model evaluation is that the only acceptable model where HGw originates from the bottom includes a closed off continental slope and extending, low-permeable horizontal clay layers, resulting in a compartmentalization of the bottom of the aquifer. When HGw originates from the top, a closed off bottom of the coastal slope is also necessary to preserve the observed amount of hypersaline groundwater since the last Glacial. This implies that regardless of HGw provenance, the interaction at the aquifer-sea connection needs to be very limited at the lower half of the aquifer to explain the observed salinities, as otherwise all HGw would have flowed out to the sea under the steep Late-Pleistocene (32 – 13.5 ka) hydraulic gradients. We can therefore limit the amount of potential combinations between lithology and HGw provenance that were posed in van Engelen et al (2018), as the current study also incorporated a steep glacial hydraulic gradient, whereas van Engelen et al (2018) kept a constant low (interglacial) hydraulic gradient.

**P17L8**: "Either of these processes"

*You mean rock salt dissolution and ...?*

For clarity we have specified these processes **[P18L14-P18L16]**:

[revised manuscript text omitted]

---

## Author Response (AR3)

**Author response to comments**

**General response**

We want to thank the reviewer for his thorough review. There were some obvious, small mistakes in the previous version of the manuscript that we clumsily overlooked, for which we apologize. We have increased our efforts to prevent this from happening again.

Replies to comments are structured as follows: First, we list the sentence to which the remark refers in quotes, preceded with **P…L…** Second comes the reviewers remark to this sentence in italics. Third comes our reply to these remarks. Finally, if relevant, how we adapted the text in red, preceded by its page and line number in the revised no markup manuscript. The markup version of the manuscript can be found at the end of this author response.

**Replies to individual comments**

**P2L10-15:** "Gossel et al. (2010) created a large-scale 3D variable density groundwater model of the Nubian Aquifer System and showed that seawater intrusion has occurred since the Pleistocene Lowstand towards the Qattara Depression (North-West Egypt). Later, Voss and Soliman (2013) showed with a parsimonious 3D model of the same groundwater system that water tables are naturally declining during the Holocene, since they receive limited recharge and are drained into oases or sabkhas. Moreover, the authors used an inventive validation method by comparing the position of discharge areas in the model with a dataset of oases or sabkha locations."

*Suggest deleting this as it is slightly off-topic in between the previous sentence and its 3 references and the more detailed discussion of these 3 references that follows*

Ok, we have removed this.

**P2L28: "**Statics"

*Just for the record, I disagree with your answer to my previous comment at this point in the manuscript. You wrote*

"*"Flow" is not a fitting term here as it does not cover the Ghyben-Herzberg approximation, which is purely hydrostatic.*"

*Hydrostatic does not imply the absence of flow, there is still flow in the Ghyben-Herzberg assumption, but the vertical head gradient in the freshwater part is ignored.*

We have removed the word "statics" from the text.

**P3L1-P3L3:** "Kashef (1983) used the Ghyben-Herzberg relationship to show that the seawater wedge reached far inland, thereby showing that the NDA should be carefully managed despite that the aquifer was gaining fresh water at that time."

*This sentence is still confusing. First of all "thereby showing" suggests that you accept his thesis, but I think the whole point of your paper is that you reject the idea of a steady-interface. Moreover, you seem to be withholding some key information because you say that at the time the aquifer was gaining fresh water. So at present this is no longer the case? Why not? Has pumping increased, has recharge gone down, or is there seawater intrusion by sea level rise? The way you formulate the sentence raises many*

*more questions than needed. I think you should just state that Kashef presented a model that assumed a steady interface. Period.*

This explanation clarified your earlier point, and we agree with you now. Changed to:

[P2L26-P2L28]: Kashef (1983) used the Ghyben-Herzberg relationship to show that the seawater wedge reached far inland.

**P3L16**: "(Sefelnasr and Sherif, 2014; Sherif et al., 1988)"

*Reference is repeated. Please pay attention to detail, there are 6 authors on this paper, how come no one picked this up?*

We apologize for this sloppy inclusion.

**P8L2:** "eastwards"

*was the delta prograding towards the east, or are you still talking about the eastern side of the delta? I assume the latter, but note that you have 3x eastwards in your sentence here.*

This is indeed a confusing sentence, changed to:

[P7L19 – P7L24]: In stress period 5, progradation started in the east, leading to the symmetrical shape in stress period 6. This progradation was caused by the decreased hydraulic gradient that led to finer sediments being deposited, which were subsequently transported eastwards in accordance with the flow direction of sea currents. During stress period 7, humans converted most river branches to a system of irrigation channels, leaving only the Rosetta and Damietta river branches to persist.

**P8L14-L15**: "This meant that for the Late-Pleistocene (stress periods 1 and 2) we had to resort to the present-day bathymetry"

*For all stress periods I assume, not just 1 and 2?*

It is true that we used to the present-day bathymetry in all stress periods to define the model geometry, but that is not the aim of this sentence. We tried to explain here that in the absence of palaeogeographic maps that define the location of the coastline (in SP1 of 2), we use the eustatic palaeo sea-level curve and the bathymetry to determine a coastline. The other stress periods have their coastlines defined by the palaeogeographic maps. We clarified the text by focusing more on the coastline and removing the word "contemporary" to refer to something occurring during a stress period. "Contemporary" can both mean "modern" and "occurring at the same time", which is quite confusing in a palaeo context.

[P7L30-P8L1]: Sea boundary cells were placed on the edge of the coastal shelf and slope, following the present-day bathymetry (GEBCO, 2014) up to the palaeocoastline. This coastline was specified, when possible, with palaeogeographic maps (Fig. 3), which meant that for the Late-Pleistocene (stress periods 1 and 2) we had to resort to using the palaeo sea-level and the present-day bathymetry to approximate the coastline.

**P8L23:** "so that its pressure is corrected for salinity"

*I assume you did the same for the sea boundary cells described above for stress periods 1 and 2?*

Yes, this was indeed redundant information, so we have removed this sub clause.

**P9L10:** "equal to the present-day average precipitation near Alexandria (WMO, 2006)"

*But how much of this becomes recharge?*

We do not know, but it shows that this recharge is comparatively high according to the present-day recharge. Clarified in text:

[P8L26-P8L32]: No recharge estimates for the dune and beach areas were available, so these areas were assigned a fixed recharge of 200 mm a$^{-1}$, equal to the present-day average precipitation near Alexandria (WMO, 2006) and the current recharge along the Levant coast (Yechieli et al., 2010). Despite being higher than present-day recharge, it is considered reasonable, as the climate was predominantly wetter throughout the Holocene than at present (Geirnaert and Laeven, 1992). This recharge allowed for the formation of freshwater lenses underneath dunes and beaches, which were observed in the area during ancient times (Post et al., 2018).

**P11L19:** "1E-04"

*Change to 0.0001*

*Seems like a tiny percentage so just to be sure, you did not mean 0.01%?*

We have changed the format of this value, but yes, we meant this. We have to be strict (1E-04%) as in this large-scale aquifer system, a small relative volume change may still indicate that the fresh-salt interface is still moving.

[P11L3]: 0.0001

**P13L15:** "T, his"

*Attention to detail please!*

Our apologies, we overlooked this in the switch from "show markup" to "show no markup".

**P13L20:** "skill"

*Can a scenario have a skill?*

We have changed the word to "error", following an earlier suggestion made by the reviewer.

[P12L30]: error

**P14L6:** "any"

*"any" is wrong here, there was some (as per your K values) but it was just much less than in the scenarios with clay.*

You are right, changed to:

[P13L6]: insignificant

**P14L22-P14L23**: "decreases the spread from 74% to 32%"

*These percentages cannot be directly inferred from Table 5. Not sure what you mean here by 'spread', so please rephrase.*

"Spread" is a general statistical term for the extent to which a distribution is stretched or squeezed. https://www.sciencedirect.com/topics/mathematics/statistical-dispersion. It is not very specific, and furthermore we noticed that we erroneously used the word "variance" previously in this paragraph which we also corrected. Say $V_{tot}$ is the array of acceptable total fresh groundwater volumes (of length 5), then we calculated these percentages by taking:

$$\frac{max(V_{tot}) - min(V_{tot})}{min(V_{tot})}$$

This was chosen to stay consistent with a previous sentence in this paragraph: "with the C-M-B-P having 74% more FGw than C-N-T-P". However, it leads to confusing text and is explained nowhere in the paper, so instead we changed it to the *range* $(max(V_{tot}) - min(V_{tot}))$, which is a generally used statistic.

[P13L21-P13L22]: This table shows that these parts of the model are the most uncertain as well, since disregarding potential deep and offshore fresh groundwater volumes decreases the range from 1133 to 478 km$^3$.

**P16L23:** "This is the zone that we expect to slowly freshen from our comparison between palaeohydrogeological and steady-state models (Fig. 9). It was however difficult to compare the observed freshening with model results directly, since the time scale over which this observed freshening has developed is often unknown (Stuyfzand, 2008)."

*What do your concentration vs time curves show in this area*

From our model scenarios, we would estimate the period of freshening to be 3000 years (Figure A4), starting with the marine regression in the east of the Delta in our boundary conditions (see Figure 3 in paper). We have added to the text:

[P15L24–P15L26]: It was however difficult to compare the observed freshening with model results directly, since the time scale over which this observed freshening has developed is often unknown (Stuyfzand, 2008). In our model simulations, freshening has continued for the last 3000 years (Fig A4). Our simulations also show that it is indeed possible that the NDA was predominantly freshened with surface water (Fig. 5), as hypothesized earlier by Geirnaert and Laeven (1992).

And we have added the following to the appendix:

[Figure]

[P43L1-P43L3]: Figure A4: Mean modelled salinity during the Holocene. The dash type indicates the scenario. Salinities were sampled in the model domain at 40 locations where a freshening was observed in the field and averaged.

**P17L15:** "originates from the bottom includes a closed off continental slope and extending, low-permeable horizontal clay layers, resulting in a compartmentalization of the bottom of the aquifer."

*This sentence does not flow, something is wrong here.*

You are right, improved as follows:

[P16L14-P16L17]: Apparent from the model evaluation is that the only acceptable model where HGw originates from the bottom has a strongly compartimentalized bottom of the aquifer, which requires a closed off continental slope and extending, low-permeable horizontal clay layers.

**P18L4:** "The most prominently simplified process"

*I was also expecting a few words here about the representation of the rivers between the apex and the coast… Given what you wrote previously, the river stages seem quite uncertain and this may have had quite some impact on groundwater flow rates.*

We have added the following text:

[P17L23-P17L28]: Furthermore, the palaeo river stages we assigned to the delta were modelled as simple as possible, namely a linear river stage profile from a static apex to coast. As mentioned earlier, this apex was in reality not static but has moved over 60 km upstream (Bunbury, 2013), which, when taken into account, would result in a lower hydraulic gradient in our model. A detailed river stage reconstruction was outside the scope of this paper. Regardless, this uncertainty in the hydraulic gradient is small compared to the big changes in the hydraulic gradient that occurred at the start of the Holocene, so we think this simplification does not impact our conclusions.

**P19L18:** "Therefore, using steady-state boundary conditions is likely to result in erroneous fresh-salt distributions and an underestimation of the uncertainty in FGw volumes."

*This has more or less been said already. Perhaps a more generic closing sentence or providing an outlook for future research efforts would form a nicer ending.*

A good suggestion. We now end the conclusion with:

[revised manuscript text omitted]

---

## Author Response (AR4)

Dear editor,

Thank you for your interest and handling our manuscript. We have not added the full track-changed manuscript at the end of this file, since there are just three minor comments.

*\* Comment 3 of the reviewer (P3L1-P3L3...): One could reconsider the word "to show". If I understand the comment of the reviewer correctly, he means that one could see the use of the Ghyphen-Herzberg relation as a contradiction to your work, as you conclude that steady state solutions are not sufficient. An alternative would be "to deomonstrate" or something else a bit weaker.*

We have followed your suggestion and changed "to show" to "to demonstrate".

*\* Comment 5 of the reviewer (P8L2...), 'to the symmetrical shape in stress period 6': Symmertical shape of what?*

Good point, we have changed this to "to the symmetrical shape of the delta in stress period 6"

*\* Comment 12 of the reviewer (P14L6...): As a suggestion, you could write "insignificant compared to the cases with clay" or something like this. An insignificant hydraulic conductivity as such would strictly speaking lead to free flow (if it is meant in the way that the resistance is insignificant), as one could interpret it in the way that there is no resistance.*

Good point. We have adapted the subclause "due to a lack of clay layers" to "compared to the scenarios with clay" and changed the word "insignificant" to "less".